# Metalearning Continual Learning Algorithms

**Kazuki Irie**                                                           *kirie@g.harvard.edu*
*Harvard University, Cambridge, MA, USA*

**Róbert Csordás**                                                       *rcsordas@stanford.edu*
*Stanford University, Stanford, CA, USA*

**Jürgen Schmidhuber**                                                   *juergen@idsia.ch*
*Center for Generative AI, KAUST, Thuwal, Saudi Arabia*
*The Swiss AI Lab, IDSIA, USI & SUPSI, Lugano, Switzerland*

**Reviewed on OpenReview:** *https://openreview.net/forum?id=IaUh7CSD3k*

## Abstract

General-purpose learning systems should improve themselves in open-ended fashion in ever-changing environments. Conventional learning algorithms for neural networks, however, suffer from catastrophic forgetting (CF), i.e., previously acquired skills are forgotten when a new task is learned. Instead of hand-crafting new algorithms for avoiding CF, we propose Automated Continual Learning (ACL) to train self-referential neural networks to metalearn their own in-context continual (meta)learning algorithms. ACL encodes continual learning (CL) desiderata—good performance on both old and new tasks—into its metalearning objectives. Our experiments demonstrate that ACL effectively resolves "in-context catastrophic forgetting," a problem that naive in-context learning algorithms suffer from; ACL-learned algorithms outperform both hand-crafted learning algorithms and popular meta-continual learning methods on the Split-MNIST benchmark in the replay-free setting, and enables continual learning of diverse tasks consisting of multiple standard image classification datasets. We also discuss the current limitations of in-context CL by comparing ACL with state-of-the-art CL methods that leverage pre-trained models. Overall, we bring several novel perspectives into the long-standing problem of CL.[1]

## 1 Introduction

Enemies of memories are other memories (Eagleman, 2020). Continually learning artificial neural networks (NNs) are memory systems in which their *weights* store memories of task-solving skills or programs, and their *learning algorithm* is responsible for memory read/write operations. Conventional learning algorithms, which are used to train NNs in the standard scenarios where all training data is available *at once*, are known to be inadequate for continual learning (CL) of multiple tasks where data for each task is available *sequentially and exclusively*, one at a time. They suffer from *catastrophic forgetting* (CF; McCloskey and Cohen (1989); Ratcliff (1990); French (1999); McClelland et al. (1995)): NNs forget, or rather, a learning algorithm erases previously acquired skills, in exchange for learning to solve a new task.

Naturally, a certain degree of forgetting is unavoidable when the memory capacity is limited, and the amount of things to remember exceeds such an upper bound. In general, however, capacity is not the fundamental cause of CF; typically, the same NNs that suffer from CF when trained sequentially on two tasks can perform well on both tasks when trained jointly on them instead (see, e.g., Hsu et al. (2018); Irie et al. (2022a)).

---

[1]This work was conducted while the authors were at the Swiss AI Lab, IDSIA (2023). An early version of this work, titled "Automating Continual Learning", was made available online in September 2023. See also closely related concurrent work by Lee et al. (2023) and Vettoruzzo et al. (2024); ours builds on our prior work on "in-context sequential multi-task learning" (Irie et al., 2022c). Our code is public: https://github.com/IDSIA/automated-cl.

The real root cause of CF lies in the learning algorithm as a memory mechanism. An effective CL algorithm should preserve previously acquired knowledge while also leveraging previous learning experiences to improve future learning, by maximally exploiting the limited memory space of model parameters. All of this is the *decision-making problem of learning algorithms.* In fact, we cannot blame conventional learning algorithms for causing CF, since they are not "aware" of such a problem. They are designed to train NNs for a given task at hand; they treat each learning experience independently (they are stationary up to certain momentum parameters in certain optimizers), and ignore any potential influence of current learning on past or future learning experiences. Effectively, more sophisticated algorithms previously proposed to combat CF (Kortge, 1990; French, 1991), such as elastic weight consolidation (Kirkpatrick et al., 2017; Schwarz et al., 2018) or synaptic intelligence (Zenke et al., 2017), often introduce manually designed constraints as regularization terms to explicitly penalize certain modifications to the previously learned model parameters/weights.

Here, instead of hand-crafting learning algorithms for continual learning, we train sequence-learning self-referential neural networks (Schmidhuber, 1992a; 1987) to metalearn their own "in-context" continual learning algorithms. We train them through gradient descent on metalearning objectives that reflect desiderata of CL—good performance on both old and new tasks. In fact, by extending the standard settings of few-shot/metalearning based on sequence-processing NNs (Hochreiter et al. (2001); Younger et al. (1999); Cotter and Conwell (1991; 1990); Mishra et al. (2018); see Sec. 2.2), the continual learning problem can also be formulated as a long-span sequence processing task (Irie et al., 2022c). We can obtain such CL sequences by concatenating multiple few-shot/metalearning "episodes," where each episode is a sequence of input/target examples corresponding to a task to be learned. As we'll see in Sec. 3, this setting also allows us to seamlessly incorporate classic desiderata of CL into the objective functions of the metalearner.

Once formulated as a sequence learning problem, we let gradient descent search for CL algorithms achieving the desired CL behaviors in the program space of NN weights. In principle, all typical challenges of CL—such as the stability-plasticity dilemma (Grossberg, 1982; Elsayed and Mahmood, 2024)—are automatically discovered and handled by the gradient-based program search process. Once meta-trained, CL is automated through recursive self-modification dynamics of the NN, without requiring any human intervention such as adding extra regularization or tuning hyper-parameters. Therefore, we call our method, Automated Continual Learning (ACL).

Our experiments focus on supervised image classification, making use of standard few-shot learning datasets for meta-training, namely, Mini-ImageNet (Vinyals et al., 2016; Ravi and Larochelle, 2017), Omniglot (Lake et al., 2015), and FC100 (Oreshkin et al., 2018), while we also meta-test on other datasets including MNIST (LeCun et al., 1998), FashionMNIST (Xiao et al., 2017) and CIFAR-10 (Krizhevsky, 2009).

Our experiments reveal various facets of in-context CL: (1) we show that without ACL, naive in-context learners suffer from "in-context catastrophic forgetting" (Sec. 4.1); we illustrate its emergence (Sec. 4.2) using comprehensible two-task settings, (2) we show very promising practical results of ACL by successfully metalearning a CL algorithm that outperforms hand-crafted learning algorithms and prior meta-continual learning methods (Javed and White, 2019; Beaulieu et al., 2020; Banayeeanzade et al., 2021) on the classic Split-MNIST benchmark (Hsu et al. (2018); Van de Ven and Tolias (2018b); Sec. 4.3), and (3) we highlight the current limitations and the need for further scaling up ACL, through a comparison with the prompt-based CL methods (Wang et al., 2022b;a) that leverage pre-trained models, using Split-CIFAR100 and 5-datasets (Ebrahimi et al., 2020).

## 2 Background

Here we provide a brief review of background concepts essential for describing our method in Sec. 3: continual learning and its desiderata (Sec. 2.1), few-shot/metalearning via sequence processing (Sec. 2.2), and linear transformer/fast weight programmer architectures (Sec. 2.3) which form the foundations of the sequence processing neural network we use in our experiments.

### 2.1 Continual Learning

The main scope of this work is continual learning (Ring, 1994; Caruana, 1997; Thrun, 1998) in *supervised* learning settings, even though high-level principles we discuss here also transfer to reinforcement learning. In addition, we focus on CL methods that keep model sizes constant (unlike certain CL methods that

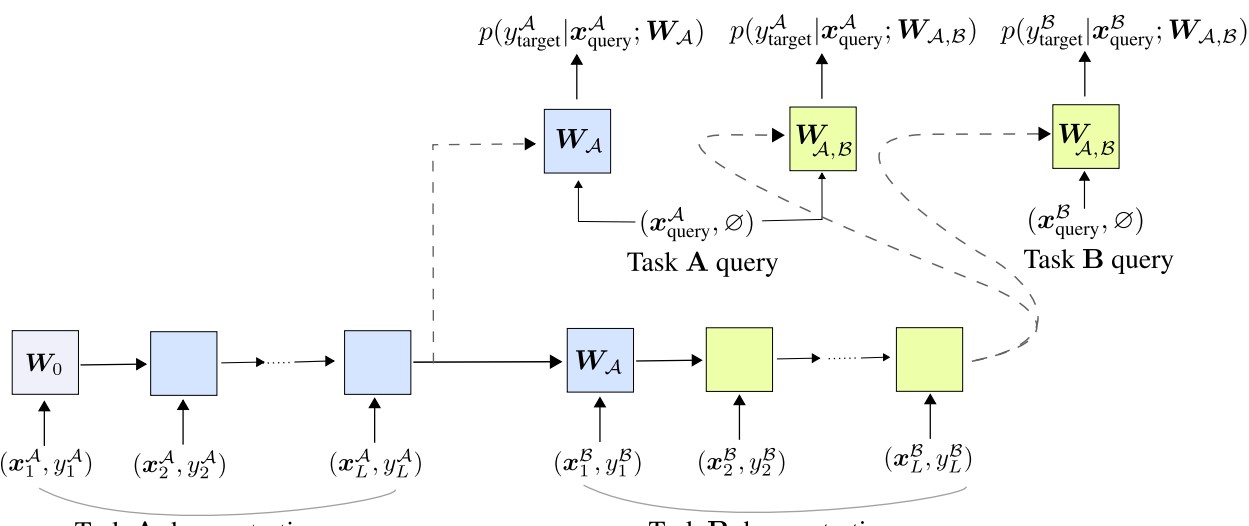

Figure 1: An illustration of sequence processing in Automated Continual Learning (ACL) using a self-referential weight matrix. The model processes a sequence of task demonstrations (i.e., x/y or input/output pairs corresponding to the task, e.g., *training* images and their labels for image classification tasks) and updates its own weight matrix (whose initial state is denoted by $W_0$) as a function of the demo sequence. We denote by $W_{\mathcal{A}}$, the weight matrix obtained after observing the sequence of Task A demonstrations (*blue*), and by $W_{\mathcal{A},\mathcal{B}}$, the matrix obtained after observing examples of Task A *then* Task B sequentially (*green*). This scheme is the same during meta-training and meta-testing. The weight matrices obtained at the task boundaries are used for evaluation: $W_{\mathcal{A}}$ and a query of Task A (e.g., a *test* image in the image classification case) are used to predict the target corresponding to the query (e.g., the label corresponding to the test image); and $W_{\mathcal{A},\mathcal{B}}$ is used to make a prediction on a query for Task A (backward transfer) *and* for Task B (forward transfer). During meta-training, the model parameters ($W_0$ in this example) are modified to optimize all such predictions using (a memory-efficient implementation of) backpropagation through time.

incrementally add more parameters as more tasks are presented; see, e.g., Rusu et al. (2016)), and we do not make use of any external replay memory (used in other CL methods; see, e.g., Robins (1995); Shin et al. (2017); Rolnick et al. (2019); Riemer et al. (2019); Zhang et al. (2022)).

Classic desiderata for a CL system (see, e.g., Lopez-Paz and Ranzato (2017); Veniat et al. (2021)) are typically summarized as good performance on three metrics: *classification accuracies* on each dataset (their average), *backward transfer* (i.e., impact of learning a new task on the model's performance on previous tasks; e.g., catastrophic forgetting is a negative backward transfer), and *forward transfer* (impact of learning the current task on the model's performance in a future task). From a broader perspective of metalearning systems, we may also want to measure *learning acceleration* (i.e., whether the system leverages previous learning experiences to accelerate future learning); here our primary focus is the classic CL metrics above.

## 2.2 Metalearning via Sequence Learning a.k.a. In-Context Learning

In Sec. 3, we formulate continual learning as a long-span sequence processing task. This is a direct extension of the classic formulation of few-shot/metalearning as a sequence learning problem, which we briefly review here.

Unlike standard learning whose goal is to train a model on a fixed task, metalearning involves training a model on many tasks or *learning episodes*, where each task serves as a learning example for the model's metalearning, so that the model learns its own algorithm to learn a new task. Each episode consists of a sequence of training examples (or *demonstrations*), followed by a test example (or *query*) whose label (or *target*) is what the model is tasked to predict; such a sequence can be processed by a sequence processing neural network. More formally, let $d$, $N$, $K$, $P$ be positive integers. Here we assume that each task is a $N$-way classification task with $K$ demonstrations for each class (the so-called $N$-way $K$-shot classification settings). At each time step $t \in \{1, ..., N \cdot K\}$,

a sequence processing NN with a parameter vector $\theta \in \mathbb{R}^P$ observes a pair $(\boldsymbol{x}_t, y_t)$ where $\boldsymbol{x}_t \in \mathbb{R}^d$ is the observation/data and $y_t \in \{1, ..., N\}$ is its label. After presentation of these $N \cdot K$ examples (demonstrations consisting of $K$ examples for each one of $N$ classes), one extra input $\boldsymbol{x} \in \mathbb{R}^d$ (a query) is fed to the model without its true label but with an "unknown label" token $\varnothing$ (thus, the model can accept up to $N+1$ different labels). The model is meta-trained to predict its true label (a target); that is, the model parameters $\theta$ are optimized to maximize the probability $p(y|(\boldsymbol{x}_1, y_1), ..., (\boldsymbol{x}_{N \cdot K}, y_{N \cdot K}), (\boldsymbol{x}, \varnothing); \theta)$ of the correct label $y \in \{1, ..., N\}$ of the input query $\boldsymbol{x}$.

Meta-training requires many such sequences/episodes, which can be constructed by using a regular dataset with $C$ classes; for each sequence, we sample $N$ random but distinct classes out of $C$ ($N < C$). The resulting classes are re-labelled such that each class is assigned to one out of $N$ distinct random label index which is unique to the sequence. For each of these $N$ classes, we sample $K$ examples. We randomly order these $N * K$ examples to obtain a unique demonstration sequence. Since class-to-label associations are randomized and unique to each sequence $((\boldsymbol{x}_1, y_1), ..., (\boldsymbol{x}_{N \cdot K}, y_{N \cdot K}), (\boldsymbol{x}, \varnothing))$, each such a sequence represents a new learning example to meta-train the model. To be more specific, this is the *synchronous* label setting of Mishra et al. (2018) where the learning phase (during which the model observes demo examples, $(\boldsymbol{x}_1, y_1)$, etc.) is separated from the prediction phase (predicting label $y$ given $(\boldsymbol{x}, \varnothing)$). We opted for this variant in our experiments as we empirically found this (at least in our specific settings) more stable than the *delayed* label setting (Hochreiter et al., 2001) where the model has to make a prediction for every input, and the label is fed to the model with a delay of one time step. The process for meta-testing is the same, except that $\theta$ is not updated.

Note that this formulation of metalearning as sequence processing is not new. Since the seminal works (Cotter and Conwell, 1990; 1991; Younger et al., 1999; Hochreiter et al., 2001), many sequence processing neural networks (see, e.g., Bosc (2015); Santoro et al. (2016); Duan et al. (2016); Wang et al. (2017); Munkhdalai and Yu (2017); Munkhdalai and Trischler (2018); Miconi et al. (2018; 2019); Munkhdalai et al. (2019); Kirsch and Schmidhuber (2021); Sandler et al. (2021); Huisman et al. (2023), including Transformers (Vaswani et al., 2017; Mishra et al., 2018)) have been trained as a metalearner (Schmidhuber, 1987; 1992a) that metalearn to learn by observing sequences of training examples (i.e., pairs of inputs and their labels). More recently, this was rebranded as *in-context learning* in the context of language modeling (Brown et al., 2020).

## 2.3 Self-Referential Weight Matrices and Recursive Self-Transformers

**General description.** Our method (Sec. 3) can be applied to any sequence-processing NN architectures in principle. Nevertheless, certain architectures naturally fit better to parameterize a self-improving continual learner. Here we use the *modern self-referential weight matrix* (SRWM; Irie et al. (2022c; 2023)) to build a generic self-modifying NN. An SRWM is a weight matrix that sequentially modifies itself as a response to a stream of input observations (Schmidhuber, 1992a; 1993).

The modern SRWM belongs to the family of linear Transformers (LTs) a.k.a. Fast Weight Programmers (FWPs; Schmidhuber (1991; 1992b); Katharopoulos et al. (2020); Choromanski et al. (2021); Peng et al. (2021); Schlag et al. (2021); Irie et al. (2021a)). Linear Transformers and FWPs are an important class of the now popular Transformers (Vaswani et al., 2017); unlike the standard ones whose computational requirements grow quadratically and whose state size grows linearly with the sequence length, LTs/FWPs' complexity is linear and the state size is constant w.r.t. sequence length, similar to the standard recurrent neural network. This property is particularly relevant for in-context CL, as we ultimately aim for such a system to continue learning over an arbitrarily long, lifelong timespan. Moreover, the duality between linear attention and FWPs (Schlag et al., 2021)—and likewise, between linear attention and gradient descent-trained linear layers (Irie et al., 2022a; Aizerman et al., 1964)—have played a key role in intuitively conceptualizing in-context learning capabilities of Transformers (von Oswald et al., 2023a; Dai et al., 2023).

The dynamics of an SRWM layer (Irie et al., 2022c) are described as follows. Let $d_{\text{in}}$, $d_{\text{out}}$, $t$ be positive integers, and $\otimes$ denote outer product. At each time step $t$, an SRWM $\boldsymbol{W}_{t-1} \in \mathbb{R}^{(d_{\text{out}}+2*d_{\text{in}}+1) \times d_{\text{in}}}$ observes an input $\boldsymbol{u}_t \in \mathbb{R}^{d_{\text{in}}}$, and outputs $\boldsymbol{o}_t \in \mathbb{R}^{d_{\text{out}}}$, while also updating itself from $\boldsymbol{W}_{t-1}$ to $\boldsymbol{W}_t$ as:

$$[\boldsymbol{o}_t, \boldsymbol{k}_t, \boldsymbol{q}_t, \beta_t] = \boldsymbol{W}_{t-1}\boldsymbol{u}_t \tag{1}$$

$$\boldsymbol{v}_t = \boldsymbol{W}_{t-1}\phi(\boldsymbol{q}_t); \quad \bar{\boldsymbol{v}}_t = \boldsymbol{W}_{t-1}\phi(\boldsymbol{k}_t) \tag{2}$$

$$\boldsymbol{W}_t = \boldsymbol{W}_{t-1} + \sigma(\beta_t)(\boldsymbol{v}_t - \bar{\boldsymbol{v}}_t) \otimes \phi(\boldsymbol{k}_t) \tag{3}$$

where $\boldsymbol{v}_t, \bar{\boldsymbol{v}}_t \in \mathbb{R}^{(d_{\text{out}}+2*d_{\text{in}}+1)}$ are value vectors, $\boldsymbol{q}_t \in \mathbb{R}^{d_{\text{in}}}$ and $\boldsymbol{k}_t \in \mathbb{R}^{d_{\text{in}}}$ are query and key vectors, and $\sigma(\beta_t) \in \mathbb{R}$ is the learning rate. $\sigma$ and $\phi$ denote sigmoid and softmax functions respectively. $\phi$ is typically also applied to $\boldsymbol{u}_t$ in Eq. 1; here we follow Irie et al. (2022c)'s few-shot image classification setting, and use the variant without it. Eq. 3 corresponds to a rank-one update of the SRWM, from $\boldsymbol{W}_{t-1}$ to $\boldsymbol{W}_t$, through the *delta learning rule* (Widrow and Hoff, 1960; Schlag et al., 2021) where the self-generated patterns, $\boldsymbol{v}_t$, $\phi(\boldsymbol{k}_t)$, and $\sigma(\beta_t)$, play the role of *target*, *input*, and *learning rate* of the learning rule respectively. The delta rule is crucial for the performance of LTs (Schlag et al., 2021; Irie et al., 2021a; 2022b; Irie and Schmidhuber, 2023b; Yang et al., 2024b;a).

The initial weight matrix $\boldsymbol{W}_0$ is the only set of trainable parameters of this layer, which encodes the seed self-modification algorithm. We use the multi-head version of the computation above (Irie et al., 2022c) to directly replace the multi-head self-attention layer in the Transformer, yielding a "Recursive Self-Transformer".

**4-learning-rate version.** In practice, we use the "4-learning rate" version (Irie et al., 2022c) of SRWM that learns to use different learning rates for each of "o", "k", "q", "$\beta$" sub-blocks of $\boldsymbol{W}_{t-1}$ by splitting $\boldsymbol{W}_{t-1}$ into sub-matrices: $\boldsymbol{W}_{t-1} = [\boldsymbol{W}_{t-1}^o, \boldsymbol{W}_{t-1}^k, \boldsymbol{W}_{t-1}^q, \boldsymbol{W}_{t-1}^b]$ that produce $\boldsymbol{o}_t$, $\boldsymbol{k}_t$, $\boldsymbol{q}_t$, and $\beta_t$, respectively, in Eq. 1. As we use 4 learning rates, the dimension of $\boldsymbol{W}_{t-1}$ becomes $\mathbb{R}^{(d_{\text{out}}+2*d_{\text{in}}+4)\times d_{\text{in}}}$, and $\beta_t = [\beta_t^o, \beta_t^k, \beta_t^q, \beta_t^b] \in \mathbb{R}^4$. For example, the corresponding update equations for the "o"-part $\boldsymbol{W}_{t-1}^o$ are:

$$\boldsymbol{o}_t^q = \boldsymbol{W}_{t-1}^o \phi(\boldsymbol{q}_t); \ \boldsymbol{o}_t^k = \boldsymbol{W}_{t-1}^o \phi(\boldsymbol{k}_t) \tag{4}$$

$$\boldsymbol{W}_t^o = \boldsymbol{W}_{t-1}^o + \sigma(\beta_t^o)(\boldsymbol{o}_t^q - \boldsymbol{o}_t^k) \otimes \phi(\boldsymbol{k}_t) \tag{5}$$

where $\boldsymbol{o}_t^q$ and $\boldsymbol{o}_t^k$ denote the "o"-part of $\boldsymbol{v}_t$ and $\bar{\boldsymbol{v}}_t$ in Eq. 2 respectively, and $\beta_t^o \in \mathbb{R}$ is one of the four learning rates that is dedicated to the "o"-part. The update equations for $\boldsymbol{W}_{t-1}^q$, $\boldsymbol{W}_{t-1}^k$, $\boldsymbol{W}_{t-1}^{\beta}$ are analogous.

## 3  Method

Here we describe the proposed approach, Automated Continual Learning (ACL).

**Problem Formalization.** Building on the formulation of *learning as sequence processing* (Sec. 2.2), we formulate continual learning also as a long-span sequence learning task. Let $D$, $N$, $K$, $L$ denote positive integers. Consider two $N$-way classification tasks **A** and **B** to be learned sequentially (this can be straightforwardly extended to more tasks). We denote the respective training datasets as $\mathcal{A}$ and $\mathcal{B}$, and test sets as $\mathcal{A}'$ and $\mathcal{B}'$. The formulation here applies to both "meta-training" and "meta-test" phases (see Appendix A.1 for more on this terminology). We assume that each datapoint in these datasets consists of one input feature $\boldsymbol{x} \in \mathbb{R}^D$ of dimension $D$ (generically denoted as vector $\boldsymbol{x}$, but it is an image in all our experiments) and one label $y \in \{1, ..., N\}$. We consider two sequences of $L$ training examples $\left((\boldsymbol{x}_1^{\mathcal{A}}, y_1^{\mathcal{A}}), ..., (\boldsymbol{x}_L^{\mathcal{A}}, y_L^{\mathcal{A}})\right)$ and $\left((\boldsymbol{x}_1^{\mathcal{B}}, y_1^{\mathcal{B}}), ..., (\boldsymbol{x}_L^{\mathcal{B}}, y_L^{\mathcal{B}})\right)$ sampled from the respective training sets $\mathcal{A}$ and $\mathcal{B}$. In practice, $L = NK$ where $K$ is the number of training examples for each class. By concatenating these two sequences, we obtain one long sequence representing CL examples to be presented to a (left-to-right processing) auto-regressive sequence model. At the end of the sequence, the model is tasked to make predictions on test/query examples sampled from both $\mathcal{A}'$ and $\mathcal{B}'$; we assume a single query example for each task (hence, without index): $(\boldsymbol{x}^{\mathcal{A}'}, y^{\mathcal{A}'})$ and $(\boldsymbol{x}^{\mathcal{B}'}, y^{\mathcal{B}'})$ respectively; which we further denote as $(\boldsymbol{x}_{\text{query}}^{\mathcal{A}}, y_{\text{target}}^{\mathcal{A}})$ and $(\boldsymbol{x}_{\text{query}}^{\mathcal{B}}, y_{\text{target}}^{\mathcal{B}})$ for clarity.

Our model is a self-referential NN that modifies its own weight matrices as a function of input observations. To simplify the notation, we denote the *state* of our self-referential NN as a single SRWM $\boldsymbol{W}_*$ (even though the model may contain many of them in practice) where we'll replace $*$ by various symbols representing the context/inputs fed to the model. Given a sequence $\left((\boldsymbol{x}_1^{\mathcal{A}}, y_1^{\mathcal{A}}), ..., (\boldsymbol{x}_L^{\mathcal{A}}, y_L^{\mathcal{A}}), (\boldsymbol{x}_1^{\mathcal{B}}, y_1^{\mathcal{B}}), ..., (\boldsymbol{x}_L^{\mathcal{B}}, y_L^{\mathcal{B}})\right)$, the model processes one x-y pair as input at a time, from left to right, in an auto-regressive manner. Let $\boldsymbol{W}_{\mathcal{A}}$ denote the state of the SRWM that has consumed the first part of the sequence, i.e., the examples from Task **A**, $(\boldsymbol{x}_1^{\mathcal{A}}, y_1^{\mathcal{A}}), ..., (\boldsymbol{x}_L^{\mathcal{A}}, y_L^{\mathcal{A}})$, and let $\boldsymbol{W}_{\mathcal{A},\mathcal{B}}$ denote the state of the SRWM after having observed the entire sequence.

**ACL Meta-Training Objectives.** The ACL meta-training objective consists in correctly predicting the target for queries of all the tasks learned so far, at every task boundary. That is, in the case of two-task scenario described above (learning Task **A** then Task **B**), we use the weight matrix $\boldsymbol{W}_{\mathcal{A}}$ to predict the label $y_{\text{target}}^{\mathcal{A}}$ from input $(\boldsymbol{x}_{\text{query}}^{\mathcal{A}}, \varnothing)$; and we use the weight matrix $\boldsymbol{W}_{\mathcal{A},\mathcal{B}}$ to predict the label $y_{\text{target}}^{\mathcal{B}}$ from input $(\boldsymbol{x}_{\text{query}}^{\mathcal{B}}, \varnothing)$ *as well as* the label $y_{\text{target}}^{\mathcal{A}}$ from input $(\boldsymbol{x}_{\text{query}}^{\mathcal{A}}, \varnothing)$. Figure 1 provides an illustration.

Let $p(y|\boldsymbol{x};\boldsymbol{W}_*)$ denote the model's output probability for label $y \in \{1,..,N\}$ given input $\boldsymbol{x}$ and model state $\boldsymbol{W}_*$. The ACL objective can be expressed as:

$$\underset{\theta}{\text{minimize}} - \big(\log(p(y^{\mathcal{A}}_{\text{target}}|\boldsymbol{x}^{\mathcal{A}}_{\text{query}};\boldsymbol{W}_{\mathcal{A}}(\theta))) + \log(p(y^{\mathcal{B}}_{\text{target}}|\boldsymbol{x}^{\mathcal{B}}_{\text{query}};\boldsymbol{W}_{\mathcal{A},\mathcal{B}}(\theta))) + \log(p(y^{\mathcal{A}}_{\text{target}}|\boldsymbol{x}^{\mathcal{A}}_{\text{query}};\boldsymbol{W}_{\mathcal{A},\mathcal{B}}(\theta))) \big)$$
$$(6)$$

for an arbitrary meta-training sequence $\big((\boldsymbol{x}^{\mathcal{A}}_1, y^{\mathcal{A}}_1),...,(\boldsymbol{x}^{\mathcal{A}}_L, y^{\mathcal{A}}_L),(\boldsymbol{x}^{\mathcal{B}}_1, y^{\mathcal{B}}_1),...,(\boldsymbol{x}^{\mathcal{B}}_L, y^{\mathcal{B}}_L)\big)$ (extensible to mini-batches using multiple such sequences), where $\theta$ denotes the model parameters (e.g., initial weights $\boldsymbol{W}_0$ for an SRWM layer) which are trained using "memory-efficient" backpropagation through time of Irie et al. (2022c).

The ACL objective function above (Eq. 6) is simple but encapsulates desiderata of continual learning (Sec. 2.1). The last term of Eq. 6 with $p(y^{\mathcal{A}}_{\text{target}}|\boldsymbol{x}^{\mathcal{A}}_{\text{query}};\boldsymbol{W}_{\mathcal{A},\mathcal{B}})$ or schematically $\boldsymbol{p}(\mathcal{A}'|\mathcal{A},\mathcal{B})$, optimizes for *backward transfer*: (1) remembering the first task **A** after learning **B** (combatting catastrophic forgetting), and (2) leveraging learning of **B** to improve performance on the past task **A**. The second term of Eq. 6, $p(y^{\mathcal{B}}_{\text{target}}|\boldsymbol{x}^{\mathcal{B}}_{\text{query}};\boldsymbol{W}_{\mathcal{A},\mathcal{B}})$ or $\boldsymbol{p}(\mathcal{B}'|\mathcal{A},\mathcal{B})$, optimizes *forward transfer* leveraging the past learning experience of **A** to improve predictions in the second task **B**, in addition to simply learning to solve Task **B** from the corresponding demonstrations. Finally, the first term of Eq. 6 incentivizes the model to learn Task **A** as soon as Task **A** demos are observed.

**Overall Model Architecture.** As discussed in Sec. 2.3, in our model, the core sequential dynamics of CL are learned by the self-referential layers. However, as an image-processing NN, our model makes use of a vision backend: We use the "Conv-4" architecture (Vinyals et al., 2016) in all our experiments, except in the last one where we use a pre-trained vision Transformer (Dosovitskiy et al., 2021). Overall, the model takes an ($\boldsymbol{x}$/image, $y$/label)-pair as input; the image is processed through a feedforward vision NN to yield a feature vector, and the label is encoded as a one-hot vector. We concatenate these two vectors, and feed it to a linear projection layer to obtain the input ($\boldsymbol{u}_t$; Eq. 1) for the first SRWM layer. Note that this is one of the limitations of this work: more general ACL should also learn to self-modify the vision components.[2]

Another crucial architectural choice that is specific to continual/multi-task image processing is normalization layers (see also related discussion in Bronskill et al. (2020)). Typical NNs used in few-shot learning (e.g., Vinyals et al. (2016)) contain batch normalization layers (BN; Ioffe and Szegedy (2015)) in the vision backend. All our models use instance normalization (IN; Ulyanov et al. (2016)) instead of BN because in our preliminary experiments, we expectantly found IN to generalize much better than BN layers in the CL setting.

## 4 Experiments

### 4.1 Two-Task Setting: Comprehensible Study

Similar to how conventional learning algorithms suffer from catastrophic forgetting in the continual learning setting, we first show that in-context learning also suffers from "in-context catastrophic forgetting," and that the ACL method (Sec. 3) can effectively help overcome it. As a minimal case to illustrate this, we focus on the two-task "domain-incremental" CL setting (see Appendix A.1 for details regarding this standard CL terminology). We consider two scenarios for meta-training *datasets* to be used to sample *tasks*: Omniglot (Lake et al., 2015) and Mini-ImageNet (Vinyals et al., 2016; Ravi and Larochelle, 2017); or FC100 (Oreshkin et al. (2018), based on CIFAR100; Krizhevsky (2009)) and Mini-ImageNet (see Appendix A.2 for data details). The order of the datasets used to sample the two tasks within the meta-training sequences is alternated for every batch. We compare the models with and without the backward transfer term in the ACL loss (the last term in Eq. 6).

Unless otherwise indicated (e.g, later for Split-MNIST; Sec. 4.3), all tasks are configured to be a 5-way classification task. This is one of the classic configurations for few-shot learning tasks, and also allows us to evaluate the principle of ACL with reasonable computational costs—like any sequence learning-based metalearning methods, scaling up to many more classes is challenging; we further discuss this in Sec. 5. For the standard datasets such as MNIST, we split the dataset into subsets of disjoint classes (Srivastava et al., 2013):

---

[2]One "straightforward" architecture fitting the bill is the MLP-mixer architecture (Tolstikhin et al. (2021); built of several linear layers), where all linear layers are replaced by the self-referential linear layers of Sec. 2.3. While we implemented such a model, it turned out to be too slow for us to conduct corresponding experiments. Our code includes a "self-referential MLP-mixer" implementation, but for further experiments, future work on such an architecture may require a more efficient implementation.

Table 1: 5-way classification accuracies using 15 demonstrations for each class. Each row is a single model. **Bold** numbers highlight the cases where in-context catastrophic forgetting is avoided through ACL. The arrow '→' indicates the in-context task presentation order.

| Meta-Training Datasets | | | Meta-Test: Demo/Train (top) & Query/Test (bottom) | | | | | |
| --- | --- | --- | --- | --- | --- | --- | --- | --- |
| | | | A | A → B | | B | B → A | |
| Task A | Task B | ACL | A | B | A | B | A | B |
| Omniglot | Mini-ImageNet | No | 97.6 ± 0.2 | 52.8 ± 0.7 | 22.9 ± 0.7 | 52.1 ± 0.8 | 97.8 ± 0.3 | 20.4 ± 0.6 |
| | | Yes | 98.3 ± 0.2 | 54.4 ± 0.8 | **98.2** ± 0.2 | 54.8 ± 0.9 | 98.0 ± 0.3 | **54.6** ± 1.0 |
| FC100 | Mini-ImageNet | No | 49.7 ± 0.7 | 55.0 ± 1.0 | 21.3 ± 0.7 | 55.1 ± 0.6 | 49.9 ± 0.8 | 21.7 ± 0.8 |
| | | Yes | 53.8 ± 1.7 | 52.5 ± 1.2 | **46.2** ± 1.3 | 59.9 ± 0.7 | 45.5 ± 0.9 | **53.0** ± 0.6 |

Table 2: Similar to Table 1 above but using MNIST and CIFAR-10 (unseen domains) for meta-testing.

| Meta-Training Datasets | | | Meta-Test: Demo/Train (top) & Query/Test (bottom) | | | | | |
| --- | --- | --- | --- | --- | --- | --- | --- | --- |
| | | | MNIST | MNIST → CIFAR-10 | | CIFAR-10 | CIFAR-10 → MNIST | |
| Task A | Task B | ACL | MNIST | CIFAR-10 | MNIST | CIFAR-10 | MNIST | CIFAR-10 |
| Omniglot | Mini-ImageNet | No | 71.1 ± 4.0 | 49.4 ± 2.4 | 43.7 ± 2.3 | 51.5 ± 1.4 | 68.9 ± 4.1 | 24.9 ± 3.2 |
| | | Yes | 75.4 ± 3.0 | 50.8 ± 1.3 | **81.5** ± 2.7 | 51.6 ± 1.3 | 77.9 ± 2.3 | **51.8** ± 2.0 |
| FC100 | Mini-ImageNet | No | 60.1 ± 2.0 | 56.1 ± 2.3 | 17.2 ± 3.5 | 54.4 ± 1.7 | 58.6 ± 1.6 | 21.2 ± 3.1 |
| | | Yes | 70.0 ± 2.4 | 51.0 ± 1.0 | **68.2** ± 2.7 | 59.2 ± 1.7 | 66.9 ± 3.4 | **52.5** ± 1.3 |

for example for MNIST which is originally a 10-way classification task, we split it into two 5-way tasks, one consisting of images of class '0' to '4' ('MNIST-04'), and another one made of class '5' to '9' images ('MNIST-59'). When we refer to a dataset without specifying the class range, we refer to the first subset. Unless stated otherwise, we concatenate 15 examples from each class for each task in the context for both meta-training and meta-testing (resulting in the sequences of length 75 for each task). All images are resized to $32 \times 32$ 3-channel images, and normalized according to the original dataset statistics. We refer to Appendix A for further details.

Table 1 shows the results when the models are meta-tested on the test set of the datasets used for meta-training. We observe that for both pairs of meta-training datasets, the models meta-trained without the ACL loss *catastrophically forget* the first task after learning the second one: the accuracy on the first task is at the chance level of about 20% for 5-way classification after learning the second task in-context (see rows with "ACL/No"). In contrast, ACL-learned CL algorithms preserve the performance of the first task ("ACL/Yes"). This effect is particularly pronounced in the Omniglot/Mini-ImageNet case (involving two very different domains). Note that there is a slight performance degradation from the single task to two-task setting in the FC100/Mini-ImageNet case (Table 1, bottom block). This is not surprising as training a model that performs well on two tasks is inherently more challenging than the single-task case, depending on the specific set of tasks involved; in particular, in the domain-incremental setting where the output layer is shared between the two tasks, "similar" tasks are inherently more confusing (e.g., in certain sequences, FC100 label 1 may be similar to Mini-ImageNet label 3; while Omniglot examples are consistently very distinguishable from Mini-ImageNet examples).

Table 2 shows evaluations of the same models but using two standard datasets, 5-way MNIST and CIFAR-10, for meta-testing. Again, the ACL-trained models preserve memory of the first task after learning the second one. In the Omniglot/Mini-ImageNet case, we even observe certain positive backward tranfer effects: in particular, in the "MNIST-then-CIFAR10" continual learning case, the performance on MNIST noticeably improves after learning CIFAR10, which is also behavior encouraged by the backward term in the ACL objective.

## 4.2   Analysis: Emergence of In-Context Catastrophic Forgetting

Here we closely examine how "in-context catastrophic forgetting" emerges during meta-training of the baseline models *without* the backward transfer term (the last/third term in Eq. 6) in the ACL objective (corresponding to the **ACL/No** case in Tables 1 and 2). We focus on the Omniglot/Mini-ImageNet case, but similar trends

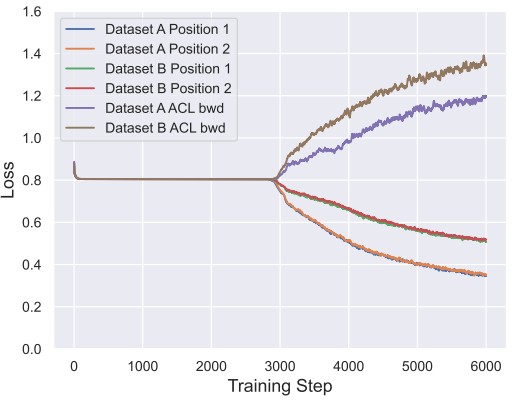
(a) Two datasets are metalearned simultaneously.

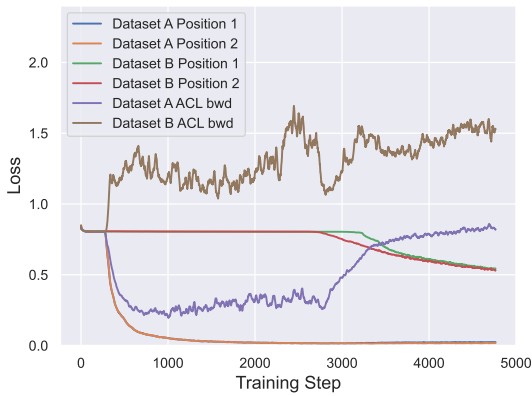
(b) One dataset is metalearned first (here Dataset A).

Figure 2: Meta-training loss terms reported separately for each dataset (A is Omniglot, B is Mini-ImageNet) and each position in the CL sequence (1 or 2) in the two-task case, yielding 6 curves. Here the ACL backward transfer terms ("*ACL bwd*" in the legend) are *not* minimized, corresponding to the **ACL/No** case in Tables 1 and 2. (a) and (b) represent two typical cases for different random seeds. In both cases, the backward transfer losses diverge (*purple* and *brown* curves) when the model "metalearns a dataset," i.e., when the model becomes capable of learning tasks sampled from the corresponding dataset (other colors), causing in-context catastrophic forgetting. Note that *blue/orange* and *green/red* curve pairs almost overlap, indicating that when the model metalearns a dataset, tasks sampled from it can be in-context learned regardless of the position.

can also be observed in the FC100/Mini-ImageNet case. Figures 2a and 2b show two representative scenarios we observe for different random seeds. These figures show an evolution of six individual meta-training loss terms (the lower the better): 4 out of 6 curves correspond to the metalearning progress reported separately for the cases where either Dataset A (here Omniglot) or B (here Mini-ImageNet) is used to sample the task at the first (1) or second (2) position in the 2-task CL meta-training demo sequences. The 2 remaining curves are the ACL backward transfer losses ("*ACL bwd*"), also reported for Datasets A and B separately.

Figure 2a shows the case where the two datasets are metalearned at the same time. We observe that when the metalearning curves go down, the backward transfer losses go up, indicating that more the model metalearns, more it tends to forget in-context. The trend is the same when one task is metalearned before the other one (Figure 2b). Here Dataset A alone is metalearned first, when B is not metalearned yet; both metalearning and backward transfer curves first go down for A—as the model has not yet metalearned the second task at this stage, nothing causes forgetting. However, at around 2,800 steps, the model also starts becoming capable of learning tasks sampled from Dataset B in-context. From this point, the backward transfer loss for Dataset A begins to go up, indicating again "opposing forces" between learning a new task and remembering a past task in-context.

These observations clearly indicate that, without explicitly including the backward transfer loss as part of the metalearning objectives, gradient descent search tends to find solutions/CL algorithms that prefer to erase previously learned knowledge. This is rather intuitive; it seems easier to find such algorithms that ignore any impact of the current learning on past learning than those that additionally preserve prior knowledge. This indicates that the ACL objective is crucial for metalearning CL algorithms that overcome catastrophic forgetting.

### 4.3 General Evaluation

**Evaluation on Standard Split-MNIST.** Here we evaluate the ACL-learned algorithms (meta-trained as described in Sec. 4.1) on the standard Split-MNIST task in both domain-incremental and class-incremental settings (Hsu et al., 2018; Van de Ven and Tolias, 2018b), and compare its performance with existing CL and meta-CL algorithms (see Appendix A.7 for the full list of references). Our comparison focuses on methods that do not rely on replay memory. Table 3 shows the results. Since the ACL models are general-purpose metalearners, they can be directly evaluated (meta-tested) on a new task, here Split-MNIST. The second-to-last row of Table 3, "ACL (Out-of-the-box model)", corresponds to our model from Sec. 4.1 meta-trained

Table 3: Classification accuracies (%) on the **Split-MNIST** domain-incremental (DIL) and class-incremental learning (CIL) settings (Hsu et al., 2018). Both tasks are 5-task CL problems. For the CIL case, we also report the 2-task case for which we can directly evaluate our out-of-the-box ACL metalearner of Sec. 4.1 (trained with a 5-way output and the 2-task ACL loss) which, however, is not applicable (N.A.) to the 5-task CIL requiring a 10-way output. Mean/std over 10 training or meta-testing runs. **No method here requires replay memory**. Non-continual joint multi-task training yields near 100% accuracy on all these tasks. See Appendix A.7 & B for further details and discussions.

| | Domain Incremental | Class Incremental | |
| Method | 5-task | 2-task | 5-task |
| --- | --- | --- | --- |
| Plain Stochastic Gradient Descent (SGD) | $63.2 \pm 0.4$ | $48.8 \pm 0.1$ | $19.5 \pm 0.1$ |
| Adam | $55.2 \pm 1.4$ | $49.7 \pm 0.1$ | $19.7 \pm 0.1$ |
| Adam + L2 | $66.0 \pm 3.7$ | $51.8 \pm 1.9$ | $22.5 \pm 1.1$ |
| Elastic Weight Consolidation (EWC) | $58.9 \pm 2.6$ | $49.7 \pm 0.1$ | $19.8 \pm 0.1$ |
| Online EWC | $57.3 \pm 1.4$ | $49.7 \pm 0.1$ | $19.8 \pm 0.1$ |
| Synaptic Intelligence (SI) | $64.8 \pm 3.1$ | $49.4 \pm 0.2$ | $19.7 \pm 0.1$ |
| Memory Aware Synapses (MAS) | $68.6 \pm 6.9$ | $49.6 \pm 0.1$ | $19.5 \pm 0.3$ |
| Learning w/o Forgetting (LwF) | $71.0 \pm 1.3$ | - | $24.2 \pm 0.3$ |
| Online-aware Meta Learning (OML), out-of-the-box | $69.9 \pm 2.8$ | $46.6 \pm 7.2$ | $24.9 \pm 4.1$ |
| + optimized number of meta-testing iterations | $73.6 \pm 5.3$ | $62.1 \pm 7.9$ | $34.2 \pm 4.6$ |
| Generative Meta-Continual Learning (GeMCL) | $63.8 \pm 3.8$ | $91.2 \pm 2.8$ | $79.0 \pm 2.1$ |
| ACL (out-of-the-box, DIL, 2-task ACL model of Sec. 4.1) | $72.2 \pm 0.9$ | $71.5 \pm 5.9$ | N.A. |
| + meta-finetuned with 5-task ACL loss, Omniglot | $\mathbf{84.5} \pm 1.6$ | $\mathbf{96.0} \pm 1.0$ | $\mathbf{84.3} \pm 1.2$ |

on Omniglot and Mini-ImageNet using the 2-task ACL objective. It performs very competitively against the best existing methods in the domain-incremental setting, while, in the 2-task class-incremental setting, it largely outperforms all of them except GeMCL, another meta-CL method. The same model can be further meta-finetuned using the 5-task version of the ACL loss (here we only used Omniglot as the meta-finetuning data). The resulting model (the last row of Table 3) outperforms all other methods in all settings studied here. We are not aware of any existing hand-crafted CL algorithms that can achieve ACL's performance without any replay memory. We refer to Appendix A.7/B for further discussions and ablation studies.

**Evaluation on diverse task domains.** Using the setting of Sec. 4.1, we also evaluate ACL models for CL involving more tasks and domains, using meta-test sequences made of MNIST, CIFAR-10, and Fashion MNIST. We also vary the number of tasks in the ACL objective: in addition to the model meta-trained on Omniglot/Mini-ImageNet (Sec. 4.1), we also meta-train a model (with the same architecture and hyper-parameters) using 3 tasks, Omniglot, Mini-ImageNet, and FC100, using the 3-task ACL objective (see Appendix A.5), resulting in meta-training that not only involves longer CL sequences but also more data. The full results of this experiment can be found in Appendix B.4. We find that both the 2-task and 3-task meta-trained models are capable of retaining the knowledge of multiple tasks during meta-testing without catastrophic forgetting; while the performance on prior tasks gradually degrades as the model learns new tasks, and performance on new tasks also becomes moderate (see also Sec. 5 on limitations). The 3-task one outperforms the 2-task one overall, encouragingly indicating a potential for further improvements even within a fixed parameter budget.

**Going beyond: limitations and outlook.** The experiments presented above effectively demonstrate that self-referential weight matrices can encode a continual learning algorithm that outperforms handcrafted learning algorithms and existing metalearning approaches for CL. While we consider this as an important result for metalearning and in-context learning in general, we note that current state-of-the-art CL methods use neither regularization-based CL algorithms nor meta-continual learning methods mentioned above, but the so-called *learning to prompt* (L2P)-family of methods (Wang et al., 2022b;a) that leverage pre-trained models, namely a vision Transformer (ViT) pre-trained on ImageNet (Dosovitskiy et al., 2021). Here we examine how ACL can leverage pre-trained models to potentially go beyond the experimental scale considered so far. To study this, we take a pre-trained (frozen) ViT model, and add self-referential layers on top of it to build a continual learner.

Table 4: Experiments with "*mini*" Split-CIFAR100 and 5-datasets tasks. Meta-training is done using **Mini-ImageNet** and **Omniglot**. Numbers marked with * are *reference* numbers (evaluated in the more challenging, original version of these tasks) which should not be directly compared to ACL performance.

|  | Split-CIFAR100 | 5-datasets |  |
|---|---|---|---|
| L2P (Wang et al., 2022a) | *83.9** ± 0.3 | *81.1** ± 0.9 |  |
| DualPrompt (Wang et al., 2022a) | *86.5** ± 0.3 | *88.1** ± 0.4 |  |
| ACL (Individual Task) | Task 1  95.9 ± 0.9 | CIFAR10 | 91.3 ± 1.2 |
|  | Task 2  85.6 ± 3.6 | MNIST | 98.9 ± 0.3 |
|  | Task 3  93.4 ± 1.4 | Fashion | 93.5 ± 2.0 |
|  | Task 4  97.0 ± 0.7 | SVHN | 66.1 ± 9.4 |
|  | Task 5  67.6 ± 7.0 | notMNIST | 76.3 ± 6.7 |
| ACL | 68.3 ± 2.0 | 61.5 ± 2.1 |  |

We use two datasets from the prior L2P work above (Wang et al., 2022b;a): 5-datasets (Ebrahimi et al., 2020) and Split-CIFAR-100 in the class-incremental setting, but we focus on our custom "*mini*" versions thereof by only using the two first classes within each task (i.e., *2-way* version); and for Split-CIFAR100, we only use the 5 first tasks (instead of 10). As we'll see, this simplified setting is enough to illustrate an important current limitation of in-context CL. Again following L2P (Wang et al., 2022b;a), we use ViT-B/16 (Dosovitskiy et al., 2021) (available via PyTorch) as the pre-trained vision model, which we keep frozen. The self-referential component uses the same configuration as in the Split-MNIST experiment. We meta-train the resulting model using Mini-ImageNet and Omniglot with the 5-task ACL loss.

Table 4 shows the results. Even in this "mini" setting, ACL's performance is far behind that of L2P methods on the original setting. Notably, the frozen ImageNet-pre-trained features with the metalearner trained on Mini-ImageNet and Omniglot are not enough to perform well on the 5-th task of Split-CIFAR100; and SVHN and notMNIST of 5-datasets; even when these tasks are evaluated in isolation. This illustrates a general limitation of in-context learning, showing the necessity for further scaling, i.e., meta-training on more diverse datasets for in-context CL and ACL to be possibly successful in more general settings.

**Ablations.** We provide several additional ablation studies in Appendix B, including those on the choice of meta-validation datasets and the effect of varying the number of in-context examples.

## 5  Discussion

**Other Limitations.**  In addition to the limitations already mentioned above, here we discuss others. First of all, as an in-context/learned learning algorithm, there are challenges in terms of both domain and length generalization. We qualitatively observe these to some extent in Sec. 4.3; further discussion and experimental results are presented in Appendix B.3 & B.5. Regarding the length generalization, we note that unlike the standard "quadratic" Transformers, linear Transformers/FWPs-like SRWMs can be trained by *carrying over states* across two consecutive batches for arbitrarily long sequences. Such an approach has been successfully applied to language modeling with FWPs (Schlag et al., 2021). This possibility, however, has not been investigated here, and is left for future work. Also, directly scaling ACL for real-world tasks involving many more classes does not seem straightforward: it would involve very long meta-training sequences. That said, it may be possible that ACL could be achieved without exactly following the process we proposed here; as we discuss below for the case of large language models (LLMs), certain real-world data may naturally give rise to an ACL-like objective. The scope of this work is also limited to image classification, which can be solved by feedforward NNs. Future work may investigate the possibility to extend ACL to continual learning of sequence learning tasks, such as continually learning new languages. Finally, ACL learns CL algorithms that are specific to the pre-specified model architecture; more general metalearning algorithms may aim at achieving learning algorithms that are applicable to any model, as is the case with many classic learning algorithms.

**Interpretability & Extracting Novel Algorithm Design Principles?** One potential application of metalearning is to discover novel learning algorithm design principles, and turn them into a human-interpretable,

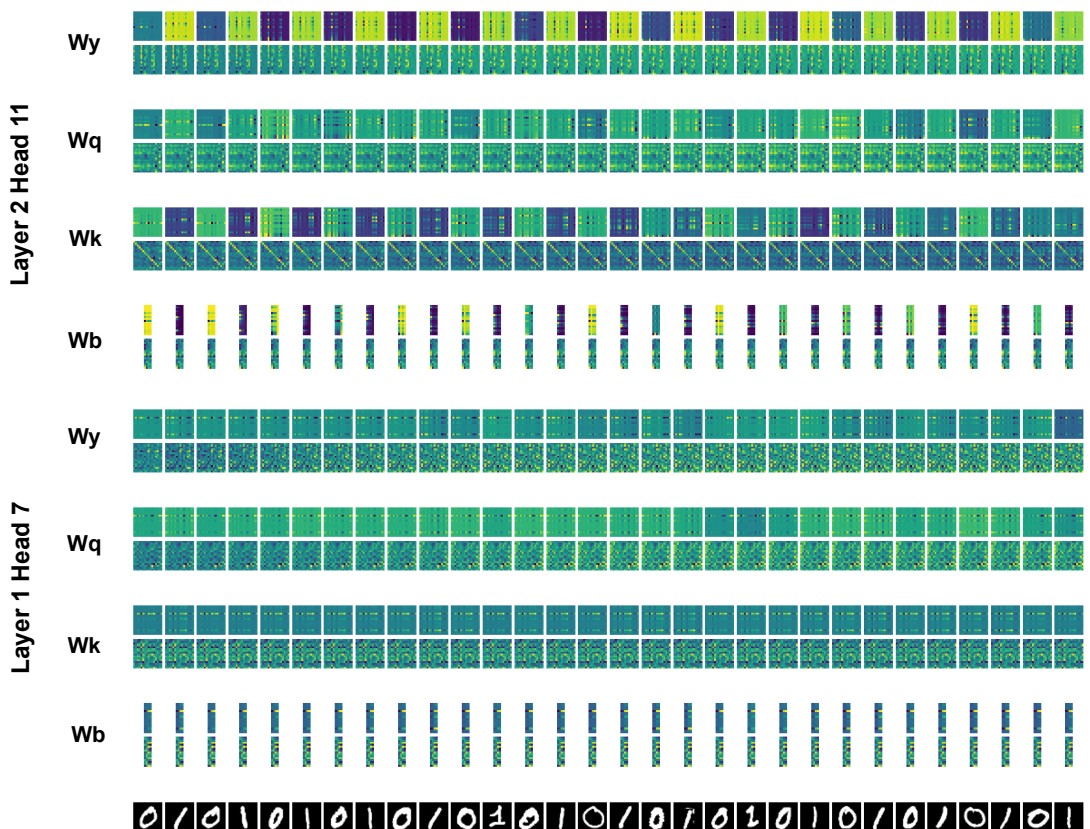

Figure 3: Visualization of weights during the presentation of **Task 1** examples.

general learning algorithm. However, as in prior work on fast weight programmers (Irie and Schmidhuber, 2023b), we found it very hard to interpret learned weight modification algorithms for CL. We provide example weight visualizations with the model used in the class-incremental setting of Split-MNIST (Sec. 4.3) while feeding meta-test demo examples to the model for two tasks from Split-MNIST (class 0 vs 1, and 2 vs 3, respectively), in Figure 3 and 4 (see Figure 5 in the appendix for presentation of the third task, 4 vs 5). One natural difficulty is to deal with a large number of weight matrices: given that our model has 2 SRWM layers with 16 heads each, and considering the 4 components of SRWM ("o", "q", "k", "$\beta$" parts; "$\beta$" part is denoted with "b"), 128 matrices have to be visualized over time (here we selected 1 head in each layer as representative examples). Future work on interpretability may need to focus on smaller models to reduce this number.

**Related work.** There are several recent works that are catagorized as meta-continual learning or continual metalearning (see, e.g., Javed and White (2019); Beaulieu et al. (2020); Caccia et al. (2020); He et al. (2019); Yap et al. (2021); Munkhdalai and Yu (2017)). For example, Javed and White (2019); Beaulieu et al. (2020) use "model-agnostic metalearning" (MAML; Finn et al. (2017); Finn and Levine (2018)) to metalearn *representations* for CL while still making use of classic learning algorithms for CL; this requires tuning of the learning rate and number of iterations for optimal performance during CL at meta-test time (see, e.g., Appendix A.7). In contrast, our approach learn *learning algorithms* in the spirit of Hochreiter et al. (2001); Younger et al. (1999); this may be categorized as "in-context continual learning." Several recent works (see, e.g., Irie and Schmidhuber (2023a); von Oswald et al. (2023b)) mention the possibility of such in-context CL but existing works (Irie et al., 2022c; Coda-Forno et al., 2023; Lee et al., 2023) that learn multiple tasks sequentially in-context do not focus on catastrophic forgetting which is one of the central challenges of CL. Here we show that in-context learning also suffers from catastrophic forgetting in general (Sec. 4.1-4.2) and propose ACL to address this problem. We also note that the use of SRWM is particularly relevant to continual metalearning. In fact, with regular linear Transformers or FWPs, the question remains regarding how to continually learn the "slow weights" (Schmidhuber, 1992b). In principle, recursive self-modification as in SRWM is an answer to this question as it

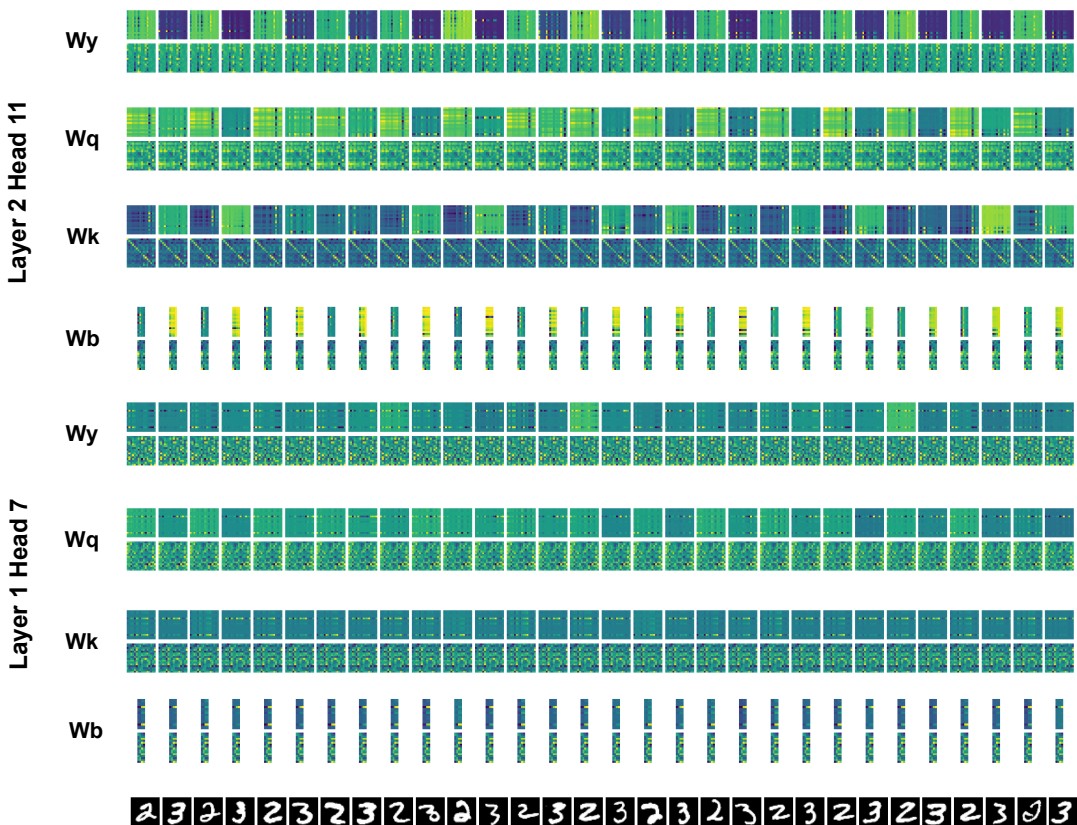

Figure 4: Visualization of weights during the presentation of **Task 2** examples.

collapses such meta-levels into a single self-referential loop (Hofstadter, 1979; Schmidhuber, 1992a). We also refer to Schmidhuber (1994; 1995); Schmidhuber et al. (1997) for other prior work on meta-continual learning.

**Artificial v. Natural ACL in Large Language Models?** Recently, the "on-the-fly" or in-context few-shot learning capability of sequence processing NNs has attracted broader interest in the context of LLMs (Brown et al., 2020). In fact, the task of language modeling itself has a form of sequence processing *with error feedback*— essential for metalearning (Schmidhuber, 1990): the correct label to be predicted is fed to the model with a delay of one time step in an auto-regressive manner (Hochreiter et al., 2001). Trained on a large amount of text covering a wide variety of credit assignment paths, LLMs exhibit certain sequential few-shot learning capabilities in practice (Brown et al., 2020). Here we explicitly/artificially constructed ACL meta-training sequences and objectives, but in modern LLMs trained on a large amount of data mixing a large diversity of dependencies using a large backpropagation span, it is conceivable that some ACL-like objectives may naturally appear in the data.

## 6 Conclusion

Our Automated Continual Learning (ACL) trains self-referential neural networks to metalearn their own in-context continual (meta)learning algorithms. ACL encodes the classic desiderata for continual learning (i.e., forward and backward transfer) into the objective function of the metalearner. ACL uses gradient descent to deal with the classic challenges of CL, to automatically discover CL algorithms with effective behavior; avoiding the need for manual, human-led algorithm design. Once trained, ACL-models autonomously run their own CL algorithms without requiring any human intervention. Our experiments reveal the problem of in-context catastrophic forgetting, and demonstrate the effectiveness of ACL to overcome it. We demonstrate promising results of ACL on the classic Split-MNIST benchmark where existing hand-crafted algorithms fail. We also highlight the need for further scaling ACL to succeed in more challenging scenarios. We believe this represents an important step towards developing open-ended continual metalearners based on neural networks.

**Acknowledgements**

This research was partially funded by ERC Advanced grant no: 742870, project AlgoRNN, and by Swiss National Science Foundation grant no: 200021_192356, project NEUSYM. We are thankful for hardware donations from NVIDIA and IBM. The resources used for this work were partially provided by Swiss National Supercomputing Centre (CSCS) projects s1145 and d123.

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

## A  Experimental Details

### A.1  Continual and Metalearning Terminologies

Here we review the classic terminologies of continual learning and metalearning used in this paper.

**Continual learning.** "Domain-incremental learning (DIL)" and "class-incremental learning (CIL)" are the two classic settings in continual learning (Van de Ven and Tolias, 2018a;b; Hsu et al., 2018). They differ as follows. Let $M$ and $N$ denote positive integers. Consider continual learning of $M$ tasks where each task is an $N$-way classification. In the DIL case, a model has an $N$-way output classification layer, i.e., the class '0' of the first task shares the same weights as the class '0' of the second task, and so on. In the CIL case, the model's output dimension is $N * M$; the class indices of different tasks are not shared, neither are the corresponding weights in the output layer. In our experiments, all CIL models have the $(N*M)$-way output from the first task (instead of progressively increasing the output size). In this work, the third variant called "task-incremental learning" which assumes that we have access to the task identity as an extra input, is not considered as it is known to make the CL problem almost trivial. CIL is typically reported to be the hardest setting among them.

**Metalearning.** Unlike standard learning, which simply involves "training" and "testing," metalearning requires introducing the "meta-training" and "meta-testing" terminologies since each of these phases involves "training/test" processes within itself. Each of them requires "training" and "test" examples. In the main text, we referred to the training examples as *demonstrations*, and the test example as consisting of a *query* (input) and a *target* (output). An alternative terminology one can find in the literature is "meta-training training/test examples", and "meta-test training/test examples" of Beaulieu et al. (2020); we opted for "demonstrations" and "queries/targets" to avoid this rather heavy terminology (except for "meta-test training iterations" of OML discussed in Appendix A.7). In both phases, a sequence-processing neural net observes a sequence of (meta-training or meta-test) training examples—each consisting of input features and a correct label, and the resulting states of the sequence processor (i.e., states of the weights in the case of SRWM) are used to make predictions on (meta-training or meta-test) test examples' input features presented to the model without the label. During the meta-training phase, we modify the trainable parameters of the metalearner through gradient descent minimizing the metalearning loss function. During meta-testing, no human-designed optimization for weight modification is used anymore; the SRWMs modify their own weights following their own learning rules defined in their forward pass (Eqs. 1-3).

## A.2 Datasets

Here we provide more details about the datasets used in this work.

For the classic image classification datasets such as MNIST (LeCun et al., 1998), CIFAR10 (Krizhevsky, 2009), and FashionMNIST (FMNIST; Xiao et al. (2017)) we refer to the original references for details.

For Omniglot (Lake et al., 2015), we use Vinyals et al. (2016)'s 1028/172/432-split for the train/validation/test set, as well as their data augmentation methods using rotation of 90, 180, and 270 degrees. Original images are grayscale hand-written characters from 50 different alphabets. There are 1632 different classes with 20 examples for each class.

Mini-ImageNet contains color images from 100 classes with 600 examples for each class. We use the standard train/valid/test class splits of 64/16/20 following Ravi and Larochelle (2017).

FC100 is based on CIFAR100 (Krizhevsky, 2009). 100 color image classes (600 images per class, each of size $32 \times 32$) are split into train/valid/test classes of 60/20/20 (Oreshkin et al., 2018).

The "5-datasets" dataset (Ebrahimi et al., 2020) consists of 5 datasets: CIFAR10, MNIST, FashionMNST, SVNH (Netzer et al., 2011), and notMNIST (Bulatov, 2011).

Split-CIFAR100 is also based on CIFAR100. The standard setting splits the original 100-way classification task into a sequence of ten 10-way classification tasks.

We use `torchmeta` (Deleu et al., 2019) which provides common experimental setups for few-shot/metalearning to sample and construct meta-train/test datasets.

## A.3 Training Details & Hyper-Parameters

We use the same model architecture and meta-training hyper-parameters in all our experiments. All hyper-parameters are summarized in Table 5. We use the Adam optimizer with the standard Transformer learning rate warmup scheduling (Vaswani et al., 2017). The vision backend is the classic 4-layer convolutional NN of Vinyals et al. (2016). Most configurations follow those of Irie et al. (2022c); except that we initialize the 'query' sub-matrix in the self-referential weight matrix using a normal distribution with a mean value of 0 and standard deviation of $0.01/\sqrt{d_{\text{head}}}$ while other sub-matrices use an std of $1/\sqrt{d_{\text{head}}}$ (motivated by the fact that a generated query vector is immediately multiplied with the same SRWM to produce a value vector). For further details, we refer readers to our public code (link provided on page 1). We conduct our experiments using a single V100-32GB, 2080-12GB or P100-16GB GPUs, and the longest single meta-training run takes about one day.

Table 5: Hyper-parameters.

| Parameters | Values |
|---|---|
| Number of SRWM layers | 2 |
| Total hidden size | 256 |
| Feedforward block multiplier | 2 |
| Number of heads | 16 |
| Batch size | 16 or 32 |

## A.4 Evaluation Procedure

For evaluation on the classic few-shot learning datasets (i.e., Omniglot, Mini-Imagenet and FC100), we use 5 different sets of 32 K random test episodes each, and report the mean and standard deviation.

For evaluation on other datasets, we use 5 different sets of randomly sampled demonstrations, and use the entire test set as the queries/targets. We report the corresponding mean and standard deviation across these 5 evaluation runs.

For the Split-MNIST (and other "Split-X") experiments, we do 10 meta-testing runs to compute the mean and standard deviation as the baseline models are also trained for 10 runs in Hsu et al. (2018); see further details in Appendix A.7.

## A.5 ACL Objectives with More Tasks

We can straightforwardly extend the 2-task version of ACL presented in Sec. 3 to more tasks. In the 3-task case (we denote the three tasks as **A**, **B**, and **C**) used in Sec. 4.3 and Appendix B.4, the objective function contains six terms. The following three terms are added to Eq. 6:

$$-\Big(\log\big(p(y^{\mathcal{C}}_{\text{target}}|\boldsymbol{x}^{\mathcal{C}}_{\text{query}};\boldsymbol{W}_{\mathcal{A},\mathcal{B},\mathcal{C}}(\theta))\big)+\log\big(p(y^{\mathcal{B}}_{\text{target}}|\boldsymbol{x}^{\mathcal{B}}_{\text{query}};\boldsymbol{W}_{\mathcal{A},\mathcal{B},\mathcal{C}}(\theta))\big)+\log\big(p(y^{\mathcal{A}}_{\text{target}}|\boldsymbol{x}^{\mathcal{A}}_{\text{query}};\boldsymbol{W}_{\mathcal{A},\mathcal{B},\mathcal{C}}(\theta))\big)\Big)$$

This also naturally extends to the 5-task loss used in the Split-MNIST experiment (Table 3), and so on. As one can observe, the number of terms quadratically increases with the number of tasks. Nevertheless, computing these loss terms isn't immediately impractical because they essentially just require forwarding the network for one step, for many independent queries. This can potentially be heavily parallelized as a batch operation. While this may still be a concern when scaling up much further, a natural open research question is whether we really need all these terms in the case we have many more tasks. Ideally, we want these models to "systematically generalize" to more tasks even when they are trained with only a handful of them (Fodor and Pylyshyn, 1988). This is an interesting research question on generalization to be studied in future work.

## A.6 Auxiliary 1-shot Learning Objective

In practice, instead of training the models only for the "15-shot learning" objective (as described in the main text, we use 15 demonstrations for each class), we also add an auxiliary loss for 1-shot learning. This incentivizes the models to learn as soon as the first demonstrations of the task become available; in practice, we found this to be generally useful for efficient meta-training.

## A.7 Details of the Split-MNIST experiment

Here we provide details of the Split-MNIST experiments presented in Sec. 4 and Table 3.

Split-MNIST is obtained by transforming the original 10-way MNIST dataset into a sequence of five 2-way classification tasks by partitioning the 10 classes into 5 groups/pairs of two classes each, in a fixed order from 0 to 9 (i.e., grouping 0/1, 2/3, 4/5, 6/7, and 8/9). Regarding the difference between domain/class-incremental settings, we refer to Appendix A.1.

For meta-finetuning using the 5-task ACL loss (last row of Table 3), we randomly sample 5 tasks from Omniglot (in principle, we should make sure that different tasks in the same sequence have no underlying class overlap; in practice, our current implementation simply randomly draws 5 independent tasks from Omniglot).

The baseline methods presented in Table 3 include: standard SGD and Adam optimizers, Adam with the L2 regularization, elastic weight consolidation (Kirkpatrick et al., 2017) and its online variant (Schwarz et al., 2018), synaptic intelligence (Zenke et al., 2017), memory aware synapses (Aljundi et al., 2018), learning without forgetting (LwF; Li and Hoiem (2016)). For these methods, we directly take the numbers reported in Hsu et al. (2018) for the 5-task domain/class-incremental settings. For the 2-task class-incremental learning case, we use Hsu et al. (2018)'s code to train the corresponding models (the number for LwF is not included as it is not implemented in their code base).

Finally we also evaluate two classic meta-CL baselines: Online-aware Meta-Learning (OML; Javed and White (2019)) and Generative Meta-Continual Learning (GeMCL; Banayeeanzade et al. (2021)). OML is a MAML-based metalearning approach. We note that as reported by Javed and White (2019) in their public GitHub code repository; after some critical bug fix, the performance of their OML matches that of a followp work by Beaulieu et al. (2020), which is a direct application of OML to another model architecture. Therefore, we focus on OML as our main MAML-based baseline. We utilize the publicly available, ready-to-use model checkpoint of Javed and White (2019) (meta-trained on Omniglot, with a 1000-way output layer).

We evaluate the corresponding OML model in two ways (Table 3). In the first, 'out-of-the-box' case, we take the meta/pre-trained model and only tune its meta-testing learning rate (which is also done by Javed and White (2019) even for meta-testing on Omniglot). We find that this approach does not perform very well on Split-MNIST (Table 3). In the other approach (denoted as 'optimized number of meta-testing iterations' in Table 3), we additionally tune the number of meta-test training iterations. We've done a grid search of the meta-test learning rate in $3 * \{1e^{-2}, 1e^{-3}, 1e^{-4}, 1e^{-5}\}$ and the number of meta-test training steps in $\{1, 2, 5, 8, 10\}$ using a meta-validation set based on an MNIST validation set (5 K held-out images from the training set); we found the learning rate of $3e^{-4}$ and meta-test training of 8 steps to consistently perform best in all our settings. We've also tried it 'with' and 'without' the standard mean/std normalization of the MNIST dataset; better performance was achieved without such normalization, which is consistent as they do not normalize the Omniglot dataset for their meta-training/testing.

The sensitivity of the MAML-based methods (Javed and White, 2019; Beaulieu et al., 2020) w.r.t. meta-test hyper-parameters has been also noted by Banayeeanzade et al. (2021). Importantly, **this is one of the characteristics of hand-crafted learning algorithms that we precisely aim to avoid using learned learning algorithms**.

OML's weak performance on the 5-task class-incremental setting is somewhat surprising, since genenralization from Omniglot to MNIST is typically straightforward in non-continual few-shot learning settings (see, e.g., Koch et al. (2015); Vinyals et al. (2016); Munkhdalai and Yu (2017)). At the same time, to the best of our knowledge, OML-trained models have not been tested in such a condition in prior work. Based on our results, it may be the case that the publicly available out-of-the-box OML model is overtuned for Omniglot/Mini-ImageNet; or the frozen "representation network" may not be ideal for genenralization.

Regarding the GeMCL baseline, we use the code and a pre-trained model (meta-trained on Omniglot) made publicly available by Banayeeanzade et al. (2021). Similarly to the ACL models, GeMCL also does not require any special tuning at meta-test time. Nevertheless, for both models, we conducted an ablation study on the effect of varying the number of meta-test training examples (5 vs. 15; 15 is the number used in meta-training). We find the consistent number, i.e., 15, to work better than 5. For the ACL version that is meta-finetuned using the 5-task ACL objective (using only the Omniglot dataset), we additionally tested the cases where we use 5 demonstrations for meta-training, and 5 or 15 for meta-testing. We find that again, the consistent number of demonstrations tends to yield the best performance. See Appendix B.3 and Table 6 for the full results.

More ablation studies can be found in Appendix B.

Table 6: Impact of the number of in-context examples. Classification accuracies (%) on **Split-MNIST** in the 2-task and 5-task class-incremental learning (CIL) settings and the 5-task domain-incremental learning (DIL) setting. For the ACL models, we use the same number of examples for meta-validation as for meta-training. According to Banayeeanzade et al. (2021), GeMCL is meta-trained with the 5-shot setting but meta-validated in the 15-shot setting.

| Number of Examples | | DIL | | CIL 2-task | | CIL 5-task | |
|---|---|---|---|---|---|---|---|
| Meta-Train/Valid | Meta-Test | GeMCL | ACL | GeMCL | ACL | GeMCL | ACL |
| 5 | 5 | - | $84.1 \pm 1.2$ | - | $93.4 \pm 1.2$ | - | $74.6 \pm 2.3$ |
| | 15 | - | $83.8 \pm 2.8$ | - | $94.3 \pm 1.9$ | - | $65.5 \pm 4.0$ |
| 15 | 5 | $62.2 \pm 5.2$ | $83.9 \pm 1.0$ | $87.3 \pm 2.5$ | $93.6 \pm 1.7$ | $71.7 \pm 2.5$ | $76.7 \pm 3.6$ |
| | 15 | $\mathbf{63.8} \pm 3.8$ | $\mathbf{84.5} \pm 1.6$ | $\mathbf{91.2} \pm 2.8$ | $\mathbf{96.0} \pm 1.0$ | $\mathbf{79.0} \pm 2.1$ | $\mathbf{84.3} \pm 1.2$ |

### A.8 Details of the Split-CIFAR100 and 5-datasets Experiment using ViT

As we described in Sec. 4.3, for the experiments on Split-CIFAR100 and 5-datasets, following Wang et al. (2022b;a), we use ViT-B/16 pre-trained on ImageNet (Dosovitskiy et al., 2021) which is available through `torchvision` (Paszke et al., 2019). In this experiments, we resize all images to $3 \times 224 \times 224$ and feed them to the ViT. We remove the output layer of the ViT, and use its 768-dimensional feature vector from the penultimate layer as the image encoding. The learnable self-referential component which is added on top of this frozen ViT encoder has the same architecture (2 layers, 16 heads) as in the rest of the paper (see all hyper-parameters in Table 5). All the ViT parameters are frozen throughout the experiment.

## B Extra Experimental Results

### B.1 Ablation Studies on the Choice of Meta-Validation Dataset

In general, when dealing with out-of-domain generalization, the choice of validation procedures to select final model checkpoints plays a crucial role in the evaluation of the corresponding method (Csordás et al., 2021; Irie et al., 2021b).

For the out-of-the-box model evaluation, the checkpoints are selected based on the average meta-validation performance on the validation set corresponding to the few-shot learning datasets used for meta-training: Omniglot and mini-ImageNet (or Omniglot, mini-ImageNet, and FC100 in the case of 3-task ACL), completely independently of the meta-test datasets used for evaluation. In contrast, in the meta-finetuning process of Table 3, we selected our model checkpoints through meta-validation on the MNIST validation dataset (we held out 5 K images from the training set).

Here we conduct an ablation study of the choice of meta-validation set, using three Split-'X' tasks where 'X' is either MNIST, FashionMNIST (FMNIST) or CIFAR-10 (in each case, we isolate 5 K images from the corresponding training set to create a validation set). In addition, we also evaluate the effect of meta-finetuning datasets (Omniglot only vs. Omniglot and mini-ImageNet).

Table 7 shows the results (we use 15 meta-training and meta-testing demonstrations, except for the Omniglot-finedtuned/MNIST-validated model from Table 3 which happens to be configured with 5 demos). Effectively, we observe that meta-validation using the validation set matching the test domain is useful. Also, meta-finetuning only on Omniglot is beneficial for the performance on MNIST when meta-validated on MNIST or FMNIST.

However, importantly, our ultimate goal is not to obtain a model that is specifically tuned for certain datasets; we aim at building models that generally work well across a wide range of tasks (ideally on any tasks); in fact, several existing works in the few-shot learning literature evaluate their methods in such settings (see, e.g., Requeima et al. (2019); Bronskill et al. (2020); Triantafillou et al. (2020)). This also goes hand-in-hand with

Table 7: Impact of the choice of meta-validation datasets. Classification accuracies (%) on three datasets: **Split-CIFAR-10**, **Split-Fashion MNIST** (Split-FMNIST), and **Split-MNIST** in the **domain-incremental** setting (we omit "Split-" in the second column). "OOB" denotes "out-of-the-box". "mImageNet" here refers to mini-ImageNet.

| Meta-Finetune Datasets | Meta-Validation Sets | Meta-Test on Split-X | | |
| --- | --- | --- | --- | --- |
| | | MNIST | FMNIST | CIFAR-10 |
| None (OOB: 2-task ACL; Sec. 4.1) | Omniglot + mImageNet | $72.2 \pm 0.9$ | $75.6 \pm 0.7$ | $65.3 \pm 1.6$ |
| Omniglot | MNIST | $\mathbf{84.3} \pm 1.2$ | $78.1 \pm 1.9$ | $55.8 \pm 1.2$ |
| | FMNIST | $81.6 \pm 1.3$ | $\mathbf{90.4} \pm 0.5$ | $59.5 \pm 2.1$ |
| | CIFAR10 | $75.2 \pm 2.3$ | $78.2 \pm 0.9$ | $\mathbf{63.4} \pm 1.4$ |
| Omniglot + mImageNet | MNIST | $\mathbf{76.6} \pm 1.4$ | $85.3 \pm 1.1$ | $66.2 \pm 1.1$ |
| | FMNIST | $73.2 \pm 2.3$ | $\mathbf{89.9} \pm 0.6$ | $66.6 \pm 0.7$ |
| | CIFAR10 | $76.3 \pm 3.0$ | $88.1 \pm 1.3$ | $\mathbf{68.6} \pm 0.5$ |

the idea of scaling up ACL (our current model is tiny; see hyper-parameters in Table 5; the vision component is also a shallow 'Conv-4' net) as well as various other considerations on self-improving continual learners (see, e.g., Schmidhuber (2018)), such as automated curriculum learning (Graves et al., 2017).

## B.2 Performance on Split-Omniglot

Here we report the performance of the ACL and GeMCL models used in the Split-MNIST experiment (Sec. 4.3) on "in-domain" 5-task 2-way Split-Omniglot. Table 8 shows the result. Performance is very similar between ACL and the baseline GeMCL on this task in the class incremental setting, unlike on Split-MNIST (Table 3) where we observe a larger performance gap between the same models. Here we also include an evaluation under the "domain incremental" setting for the sake of completeness but note that GeMCL is not originally meta-trained for this setting.

Table 8: Accuracies (%) on 5-task 2-way Split-Omniglot. Mean/std computed over 10 meta-test runs.

| Method | Domain Incremental | Class Incremental |
| --- | --- | --- |
| GeMCL | $64.6 \pm 9.2$ | $97.4 \pm 2.7$ |
| ACL | $92.3 \pm 0.4$ | $96.8 \pm 0.8$ |

## B.3 Varying the Number of In-Context Examples/Demonstrations

Table 6 shows an ablation study on the number of demonstrations used for meta-training and meta-testing on the Split-MNIST task. We observe that for the ACL model trained only with 5 examples during meta-training, providing more examples (15 examples) during meta-testing is not beneficial. In fact, it even largely degrades performance in certain cases (see the last column); this is one form of the "length generalization" problem. When the number of meta-training examples is consistent with the one used during meta-testing, the 15-example case (i.e., providing more demonstrations to the model) consistently outperforms the 5-example one.

## B.4 Varying the Number of Tasks in the ACL Meta-Training Objective

Table 9 provides the complete results discussed in Sec. 4.3 under "Evaluation on diverse task domains".

## B.5 Further Discussion on Limitations

Here we provide further discussion and experimental results on the limitations of learned learning algorithms.

Table 9: 5-way classification accuracies (%) using 15 examples for each class for each task in the context. 2-task models are meta-trained on Omniglot and Mini-ImageNet, while meta-training of the 3-task model additionally involves FC100. 'A → B' in 'Demo/Train' column indicates that models sequentially observe meta-test demo examples of a task sampled from Dataset A then those from B; evaluation is only done at the end of the sequence. "ACL No" is the baseline 2-task model that is meta-trained without the ACL loss.

| Meta-Testing | | Number of Meta-Training Tasks | | |
|---|---|---|---|---|
| Demo/Train | Query/Test | 2 (ACL No) | 2 | 3 |
| A: MNIST-04 | A | 71.1 ± 4.0 | 75.4 ± 3.0 | 89.7 ± 1.6 |
| B: CIFAR10-04 | B | 51.5 ± 1.4 | 51.6 ± 1.3 | 55.3 ± 0.9 |
| C: MNIST-59 | C | 65.9 ± 2.4 | 63.0 ± 3.3 | 76.1 ± 2.0 |
| D: FMNIST-04 | D | 52.8 ± 3.4 | 54.8 ± 1.3 | 59.2 ± 4.0 |
| | Average | 60.3 | 61.2 | 70.1 |
| A → B | A | 43.7 ± 2.3 | 81.5 ± 2.7 | 88.0 ± 2.2 |
| | B | 49.4 ± 2.4 | 50.8 ± 1.3 | 52.9 ± 1.2 |
| | Average | 46.6 | 66.1 | 70.5 |
| A → B → C | A | 26.5 ± 3.2 | 64.5 ± 6.0 | 82.2 ± 1.7 |
| | B | 32.3 ± 1.7 | 50.8 ± 1.2 | 50.3 ± 2.0 |
| | C | 56.5 ± 8.1 | 33.7 ± 2.2 | 44.3 ± 3.0 |
| | Average | 38.4 | 49.7 | 58.9 |
| A → B → C → D | A | 24.6 ± 2.7 | 64.3 ± 4.8 | 78.9 ± 2.3 |
| | B | 20.6 ± 2.3 | 47.5 ± 1.0 | 49.2 ± 1.3 |
| | C | 38.5 ± 4.4 | 32.7 ± 1.9 | 45.4 ± 3.9 |
| | D | 36.1 ± 2.5 | 31.2 ± 4.9 | 30.1 ± 5.8 |
| | Average | 30.0 | 43.9 | 50.9 |

**Domain generalization.** As a data-driven learned algorithm, the domain generalization capability is a typical limitation as it depends on the diversity of meta-training data. Certain results we presented above are representative of this limitation. In particular, in Table 7, the model meta-trained/finetuned on Omniglot using Split-MNIST as the meta-validation set does not perform well on Split-CIFAR10.

While meta-training and meta-validation on a larger/diverse set of datasets may be an immediate remedy to obtain more robust ACL models, we note that since ACL is also a "continual metalearning" algorithm (Sec. 5), an ideal ACL model should also continually incorporate and learn from more data during potentially lifelong meta-testing; we leave such an investigation for future work.

**Comment on meta-generalization.** We also note that in general, "unseen" datasets do not necessarily imply that they are harder tasks than "in-domain" test sets; when meta-trained on Omniglot and mini-ImageNet, meta-generalization on "unseen" MNIST is easier (the accuracy is higher) than on the "in-domain" test set of mini-ImageNet with heldout/unseen classes (compare Tables 1 and 2).

**Length generalization.** We qualitatively observed the limited length generalization capability in Table 9 (meta-trained with up to 3 tasks and meta-tested with up to 4 tasks) and in Appendix B.3 (meta-trained using 5 demonstrations and meta-tested using 15 demos). Similarly, we observed that the performance on Split-Omniglot in the domain-incremental setting of Sec. B.2 degraded as we increased the number of tasks: accuracies for 5, 10 and 20 tasks are $92.3\% \pm 0.4$, $82.0\% \pm 0.4$ and $67.6\% \pm 1.1$, respectively. As noted in Sec. 5, this is a general limitation of sequence processing neural networks, and there is a potential remedy for this limitation (meta-training on more tasks and with "context carry-over") which we leave for future work.

### B.6 More Visualizations

Figure 5 shows the continuation of Figures 3 and 4, corresponding to the demonstrations of the third task (class '4' vs. '5').

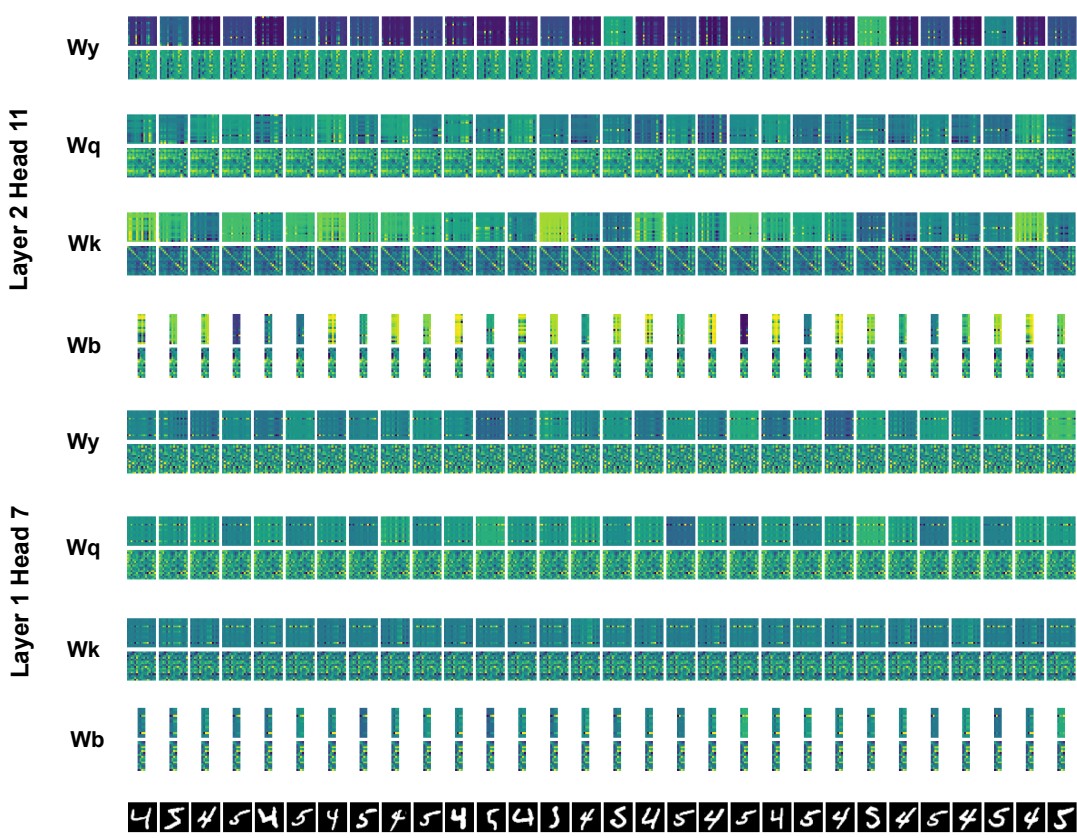

Figure 5: Visualization of weights during the presentation of **Task 3** examples.

