# OpenReview forum: "Metalearning Continual Learning Algorithms"
_TMLR — Accepted by TMLR_

### Review · Reviewer_qEUp · 2024-12-04

**Summary Of Contributions:**

The authors propose a method for `in context' continual learning. They use a previously proposed method 'self-referential weight matrix' and adapt its objective function such that it can handle multiple in-context tasks. Rather than only requiring a correct prediction for a single meta-train test example, a meta-train test example is included for every task of the meta-train sequence. This makes the SRWM learn to remember the examples that were early in the sequence. This method is tested first on two task sequences where the meta-train datatest are Mini-ImageNet and Omniglot and the meta-test sets either those two datasets or MNIST and CIFAR10. The proposed method is further compared to classical regularization methods of continual learning on a split MNIST benchmark. The limitations of the method are finally explored by testing a cifar100 benchmark and comparing to prompt based methods, all using a pretrained ViT model. An effort is made to understand what the meta-learner has learned, but this has no definitive conclusion.

**Audience:**

Yes

**Broader Impact Concerns:**

/

**Claims And Evidence:**

No

**Requested Changes:**

* (For acceptance) Improve the clarity of the paper, especially considering the training and testing setup and the model architecture
* (For acceptance) Clearly state that this assumes a set of labelled examples during meta-testing, and that the methods that are compared to do not make such assumptions, which is likely the reason that they are not performing well.
* (For acceptance) Include more related baselines that are adopted to the new problem setting, not simply ones that are taken from different settings.

**Strengths And Weaknesses:**

### Strengths

* In context continual learning may be an interesting setting. As far as I can tell this has not been explored previously, but it may be hard to find related work in this area due to the many different terminology used.
* The results in Table 1 show that the proposed method works and the model can accurately classify examples that belong to the first task in the sequence. It showcases that the SRWM technique can act as a memory that stores examples of the first part of the sequence

### Weaknesses
* The paper is hard to follow in many places. I found it especially challenging to understand how the data was set up and how a training iteration works. The information is scattered over multiple sections and the appendix, so it is difficult to get the complete picture. For instance, in Figure 1 the process of meta-training is explained, but you have to carefully read that this is just the meta-training and the testing is not included. Since there are examples that include the word 'test' , that quickly becomes confusing. A figure depicting the entire setup, including meta-training and meta-testing would be helpful. Another example is the fact that the labels are randomized every trail is not in Section 3, which is important information. The training objective mentions parameters $\theta$ to optimize, but they are nowhere in the objective itself. It requires additional information from the related work section and a future section on the model architecture to understand what $\theta$ refers to. I understand that the setup of this kind of experiments is hard, because of its meta-learning nature. Yet at this point, I think the text can be significantly improved.

* The method is proposed as a general continual learning method, that does not require rehearsal. E.g. the title: "Metalearning Continual Learning Algorithms", at the start of Section 2 ("the main focus of this work is on continual learning") and the comparison the methods like EWC, SI and MAS. I agree that this is a form of continual learning and that it can be called as such, but it requires the very specific 'in-context' assumption, that during testing there is some data available. By exclusively comparing to traditional methods, who do not make such assumption, the impression is given that this is a more general continual learning method. I believe the distinction between this method and the ones compared to should be explained better, as they were never intended to work in this setting anyway.

* Since this method proposes itself as the first to test out this setting a few simple baselines would have been appropriate. As far as I understand, a backbone model is trained, on top of which the SRWM is learned. The backbone should thus learn general image features. One easy baseline I can think of is to simply store the feature vectors of the labelled examples in a meta-test sequence. Then the distances to the meta-test test examples can be calculated, and e.g. a nearest neighbor classification to predict the correct example. In fact, it may be possible that this is what the SRWM module is learning. There are some ablations on how the method may be working, but I find those rather unsatisfying. The SRWM in itself is rather hard to understand, so it would be a lot more insightful to test some simpler ideas first and then show that the SRWM has something extra that makes it work better than some naive ideas.

---

> ### Author Response · Authors · 2024-12-11
> **Response to Reviewer qEUp**
>
> We thank the reviewer for the valuable time reviewing our work. We would like to clarify the reviewers' concerns as follows.
>
> > The paper is hard to follow in many places. I found it especially challenging to understand how the data was set up and how a training iteration works.
>
> > Another example is the fact that the labels are randomized every trail is not in Section 3, which is important information.
>
> Thank you very much for this valuable feedback. We reviewed this standard setting from the prior few-shot learning and meta-learning literature as a background material in Sec. 2.2 (second paragraph) where we also describe that *“class-to-label associations are randomized and unique to each sequence“*.
>
> > in Figure 1 the process of meta-training is explained, but you have to carefully read that this is just the meta-training and the testing is not included. Since there are examples that include the word 'test' , that quickly becomes confusing. A figure depicting the entire setup, including meta-training and meta-testing would be helpful.
>
> > (**Requested changes**)  (For acceptance) Improve the clarity of the paper, especially considering the training and testing setup and the model architecture
>
> Thank you for this suggestion. Given that there is no difference between meta-training and meta-testing on the level of details depicted in Figure 1, it also illustrates meta-testing; we will specify this in the caption in the revision.
>
> Regarding the metalearning terminology (“meta-training training/test examples”), our current terminology follows the one suggested in prior work (as explained in Appendix A.1). However, we also agree how confusing this is. We plan to change it to “demonstration” and “query” instead of “training” and “testing” to eliminate the use of “test” except in meta-”testing”.
> We believe this would resolve the root of the reviewer’s confusions. Thank you for this valuable feedback.
>
> > The training objective mentions parameters θ to optimize, but they are nowhere in the objective itself. It requires additional information from the related work section and a future section on the model architecture to understand what refers to.
>
> Thank you for pointing this out. Just below Eq. (6), we specified *“where θ denotes the model parameters (for the SRWM layer, it is the initial weights W_0)“*.
>
> That said, we agree with the reviewer that we could make this dependency more explicit in Eq. (6); In the revision we will make θ appear in the equation as: W_A(θ) and W_{A, B}(θ) instead of W_A, and W_{A, B}.
>
> > it requires the very specific 'in-context' assumption, that during testing there is some data available.
>
> > (**Requested changes**)   (For acceptance) Clearly state that this assumes a set of labelled examples during meta-testing, and that the methods that are compared to do not make such assumptions, which is likely the reason that they are not performing well.
>
> We would like to clarify that there is a misunderstanding; there is no such an assumption at meta-test time.
>
> E.g., to train/evaluate a model on MNIST, any algorithm (the conventional approach or ours) requires an MNIST training set and test set. Our method uses the training set as a **meta-testing training** set, and the test set as a **meta-testing test** set.
>
> The core difference between in-context learning and the conventional learning is the assumption on the availability of **meta-training** data (e.g., Omniglot), like any other metalearning approaches such as OML and GeMCL from Table 3, while the meta-test condition is fair compared to any conventional approaches.
>
> > By exclusively comparing to traditional methods, who do not make such assumption, the impression is given that this is a more general continual learning method.
>
> > (**Requested changes**) (For acceptance) Include more related baselines that are adopted to the new problem setting, not simply ones that are taken from different settings.
>
> There appears to be some oversight since this critique is factually inaccurate; our comparison is not *“exclusive”* to the traditional methods such as *“like EWC, SI and MAS.”* Please take another look at Table 3 that also includes two classic meta-continual learning baselines: OML and GeMCL.
>
> > One easy baseline I can think of is to simply store the feature vectors ...
>
> We believe this request becomes obsolete after resolving the misunderstanding above. The meta-continual learning has a rich literature, and there is no *“easy”* baseline that works well.  The approach the reviewer describes belongs to the family of metric-based approaches called the “Prototypical Networks” (Snell et al. NeurIPS 2017 “Prototypical Networks for Few-shot Learning”); and our baseline, GeMCL (NeurIPS 2021) is an advanced method in the same family but tailored to the meta-continual learning setting.
>
> To conclude, we thank the reviewer once again for their valuable time reviewing our work; we hope our response resolves remaining concerns and misunderstandings.

---

> > ### Comment · Reviewer_qEUp · 2025-01-08
> >
> > My apologies for not reacting sooner. Thank you for the clarifications and edits to the manuscript, I believe my concerns are sufficiently clarified (and rectified, I did miss some of the baselines in Table 3).
> >
> > I believe the manuscript is ready for acceptance.

---

> > > ### Author Response · Authors · 2025-01-08
> > > **Thank you**
> > >
> > > We understand the delayed response, considering the busy time of year when this review took place.
> > >
> > > Thank you very much for your response and the positive final recommendation.

---

### Review · Reviewer_NC5t · 2024-12-05

**Summary Of Contributions:**

This paper proposes an automated continual learning (ACL) framework that trains self-referential neural networks to meta-learn their own continual learning algorithms. By extending the concepts of in-context learning to the realm of continual learning, the authors address the problem of catastrophic forgetting and show that ACL can be a powerful alternative to hand-crafted continual learning algorithms. The main contributions include introducing the ACL objective that encodes continual learning desiderata, and empirically demonstrating its effectiveness on classic benchmarks such as Split-MNIST and other diverse datasets.

**Audience:**

Yes

**Broader Impact Concerns:**

No concerns in this regard.

**Claims And Evidence:**

Yes

**Requested Changes:**

1. Formalize the task definition for meta-learner transfer.
2. Clarify the distinction between in-context catastrophic forgetting and conventional catastrophic forgetting.
4. Add backward transfer (BWT) and forward transfer (FWT) metrics to the experiments.
5. Include the oracle joint training as a baseline.
6. Add further discussion on Figure 3's visualization.

**Strengths And Weaknesses:**

**Strengths**：

1. The paper presents an innovative approach by leveraging self-referential networks to perform continual learning without hand-crafting constraints.
2. The authors introduce the concept of 'in-context catastrophic forgetting' and demonstrate how ACL effectively mitigates this problem.
3. The empirical evaluation shows promising results, especially on Split-MNIST, where ACL outperforms existing meta-continual learning methods without replay memory.

**Weaknesses**：
1. I understand that the setup of ACL is to learn a meta-learner from the visible long sequence continual learning task and apply the learned learner to the unseen continual learning scenario. However, the latter part of the transfer is not formalized in the task definition in Section 3. If my understanding is incorrect, please clarify; otherwise, I recommend adding clear definitions to help readers understand the task scenario rather than using general terms like in-context learning.

2. The distinction between 'in-context catastrophic forgetting' (described in Section 4.2) and conventional 'catastrophic forgetting' in continual learning is not sufficiently clear. Additionally, the visual representation in Figure 2 is confusing. Specifically, the curves in Figure 2(a) show divergence around step 3000, while similar divergence is not evident in Figure 2(b). Further clarification is required.

3. The experiments report only the average accuracy after continual learning. Metrics like backward transfer (BWT) and forward transfer (FWT) should also be included to give a more comprehensive understanding of the algorithm's impact on different aspects of continual learning.

4. The oracle joint training method is missing, which is necessary to establish an upper bound on task performance.

5. Lack of further analysis and discussion of Figure 3 visualizations. The learned model of ACL is very interesting and worthy of in-depth exploration. It is suggested that the authors briefly explain the visualizations in this paper to provide insights for the continual learning community.

---

> ### Author Response · Authors · 2024-12-11
> **Response to Reviewer NC5t**
>
> We thank the reviewer for the valuable feedback. We appreciate the reviewer’s thorough understanding of our contributions. We would like to clarify the remaining concerns as follows.
>
> > 1. ... the latter part of the transfer is not formalized in the task definition in Section 3. If my understanding is incorrect, please clarify; otherwise, I recommend adding clear definitions to help readers understand the task scenario rather than using general terms like in-context learning.
>
> > (**Requested changes**) Formalize the task definition for meta-learner transfer.
>
> Thank you for this clarification question. The corresponding formalization can be found in Sec 3. first paragraph “Task Formulation” (we wrote *“The formulation here applies to both “meta-training” and “meta-test” phases“*). There is no distinction between processing of “meta-training” or “unseen meta-testing” sequences; the only difference is that during meta-training, we evaluate the meta-objective function and update the model parameters.
>
> We also would like to emphasize that our method description in Sec 3 does not use the term “in-context learning” at all. We refer to in-context learning because it is truly an instantiation of thereof, and this may accelerate understanding of familiar readers; but our description (Sec 2.2 and 3) itself does not require any pre-knowledge about it as we explain it from scratch.
>
> We hope this clarifies; but if the reviewer still thinks that we are “using general terms like in-context learning” in a way that alienates unfamiliar readers, please let us know which specific text passages need improvement; we would love to fix it!
>
> > 2. The distinction between 'in-context catastrophic forgetting' (described in Section 4.2) and conventional 'catastrophic forgetting' in continual learning is not sufficiently clear.
>
> Similarly to how the “conventional learning” (e.g., gradient descent) of a single task can be extended to “continual learning” of multiple tasks in a sequential manner, “in-context learning” of a single task can be extended to “in-context continual learning” of multiple tasks in a sequential manner. The former setting is known to suffer from “catastrophic forgetting” (dramatic loss of skills to solve the previous tasks); we show that the latter suffers from the analogous problem, which we call “in-context catastrophic forgetting”.
>
> > the visual representation in Figure 2 is confusing. Specifically, the curves in Figure 2(a) show divergence around step 3000, while similar divergence is not evident in Figure 2(b). Further clarification is required.
>
> Figure 2a and 2b are independent runs with different seeds; as such, there is no reason to expect both to start “converging” at the same number of steps. We will improve the corresponding passage in Sec 4.2 as “ Figures 2a and 2b show two representative cases we typically observe *for different random seeds.*”. Thank you very much for pointing this out.
> Regarding further clarifications, I kindly ask the reviewer to refer to our response to Reviewer 5yTC.
>
> > 3. The experiments report only the average accuracy after continual learning. Metrics like backward transfer (BWT) and forward transfer (FWT) should also be included to give a more comprehensive understanding of the algorithm's impact on different aspects of continual learning.
>
> > (**Requested changes**)  Add backward transfer (BWT) and forward transfer (FWT) metrics to the experiments.
>
> Thank you for pointing this out! Please note that we provide a much more complete picture than just BWT and FWT: we provide the performance on *every* dataset at each stage of continual learning in Tables 1 and 2 (as well as in Table 10 in the appendix). Regarding Table 3, as Split-MNIST is a standard benchmark, we follow the standard evaluation focused on average accuracy, which allows us to directly compare to numbers reported in prior work. We hope this clarifies.
>
> > 4. The oracle joint training method is missing, which is necessary to establish an upper bound on task performance.
>
> > (**Requested changes**) Include the oracle joint training as a baseline.
>
> Thank you for pointing this out! We will add the oracle to Table 3. That said, we would like to emphasize that none of our claims/conclusions would be affected as they require no comparison to such an oracle, since our main results are focused on combating catastrophic forgetting.
>
> > 5. Lack of further analysis and discussion of Figure 3 visualizations.
>
> > (**Requested changes**) Add further discussion on Figure 3's visualization.
>
> Thank you for this encouraging comment. However, as we explained in the paper (paragraph “Interpretability” under Sec 5); unfortunately, we can not state more than what we wrote. As we suggested there, future work targeting interpretability would have to use even simpler/toy tasks and focus on much smaller models.
>
> To conclude, we thank the reviewer once again for their valuable feedback; we hope our response clarifies all the remaining concerns!

---

> > ### Comment · Reviewer_NC5t · 2024-12-25
> >
> > Thank the authors for their response. After reviewing their clarifications and the revised manuscript, all my concerns have been addressed. Therefore, I recommend accepting the manuscript as is.

---

> > > ### Author Response · Authors · 2024-12-25
> > > **Thank you**
> > >
> > > Thank you very much for your response.  We extend our best wishes for a happy holiday season.

---

### Review · Reviewer_5yTC · 2024-12-10

**Summary Of Contributions:**

The authors present a method for continual learning which trains sequence-processing self-referential neural networks to meta-learn their own in-context continual meta-learning algorithms. The main contribution of this paper is to reveal the original catastrophic forgetting problem of in-context learning algorithms, and propose ACL to effectively solve such "in-context catastrophic forgetting". The authors demonstrate the superiority of ACL by comparing it with hand-crafted CL algorithms and prior meta-continual learning methods, as well as highlight its limitations by comparing to more recent prompt-based state-of-the-art CL methods.

**Audience:**

Yes

**Broader Impact Concerns:**

No major concerns.

**Claims And Evidence:**

Yes

**Requested Changes:**

- If possible, please add experiments to illustrate the applicability of the proposed method on continual reinforcement learning tasks.
- Please add additional discussion regarding the small magnitude of forgetting exhibited by the ACL-learned CL algorithms in Tables 1 and 2.
- Could you clarify the experimental setup and results analysis of Figure 2?

**Strengths And Weaknesses:**

**Strengths:**

Overall, the paper is well written and the method is well tested empirically on the Split-MNIST benchmark. The method is also, mostly, well discussed.

**Weaknesses:**

There are several limitations that should be addressed. First of all, we note that the method is a relatively simple integration of the established desiderata for continual learning under continual meta-learning framework. While the paper still has value, the methodological contribution of the paper is fairly weak. Other points:
- The authors just focus their experiments on the classical supervised image classification, and do not discuss the adaptability of the proposed method to the more challenging reinforcement learning tasks.
- In Table 1, for the FC100/Mini-ImageNet case, the ACL-learned CL algorithms still exhibit a forgetting of 6% to 7% classification accuracy on the first task (Similar results also appear in Table 2). The authors do not provide any analysis in the paper regarding this phenomenon and the main reasons for it.
- The results in Figure 2 are very confusing. Firstly, for Figure 2(a), given that meta-training is performed on the continual learning sequence consisting of two tasks, why is it also said that two tasks are learned simultaneously? Furthermore, if two tasks are to be learned simultaneously, how to set them to appear in the first or second position respectively? Isn't that a contradiction? Secondly, for Figure 2(b), given that task A is specified to be learned first, why is it still possible that task A appears in the second position and task B appears in the first position?

---

> ### Author Response · Authors · 2024-12-11
> **Response to Reviewer 5yTC**
>
> We thank the reviewer for their valuable feedback and their clear understanding of the methodology presented in our work. In particular, we thank the reviewer for highlighting the overall clarity among the strengths.
>
> > There are several limitations that should be addressed. First of all, we note that the method is a relatively simple integration of the established desiderata for continual learning under continual meta-learning framework. While the paper still has value, the methodological contribution of the paper is fairly weak.
>
> We thank the reviewer for this insightful comment. While the methodology may be a “simple” extension of a well-known framework in metalearning (actually few-shot learning via sequence learning) to continual learning, we are unaware of any prior work which identifies and discusses the “in-context catastrophic forgetting” problem as we do.
>
> Also, we show that our meta-learned CL algorithm is effectively successful at Split-MNIST unlike any hand-crafted algorithms and other prior types of meta-learning methods for CL. We are not aware of any trivial solutions for Split-MNIST even though Split-MNIST is a "toy" task compared to other larger scale tasks (we also explicitly acknowledge this in the paper). We believe this is a non-trivial achievement and a significant contribution in meta-learning research of learning algorithms.
>
> > The authors just focus their experiments on the classical supervised image classification
>
> > (**Requested changes**) If possible, please add experiments to illustrate the applicability of the proposed method on continual reinforcement learning tasks.
>
> Thank you for this excellent suggestion.  Indeed, we are highly interested in extending our framework to the reinforcement learning (RL) settings. However, RL brings its own challenges and raises other interesting questions (e.g., unless we restrict ourselves to the offline setting, an interesting question would be to explore continual improvements on the exploration ability of the agent). In our view, discussing and addressing such challenges would merit an independent paper, which we would love to investigate in the future work.
>
> > In Table 1, for the FC100/Mini-ImageNet case, the ACL-learned CL algorithms still exhibit a forgetting of 6% to 7% classification accuracy on the first task
>
> > (**Requested changes**) Please add additional discussion regarding the small magnitude of forgetting exhibited by the ACL-learned CL algorithms in Tables 1 and 2.
>
> We thank the reviewer for pointing this out. Training a model that performs well on two tasks is inherently more challenging than the single-task case, depending on the specific set of tasks involved. We note that certain task-combinations (Table 1, top block, Omniglot + Mini-ImageNet) do not exhibit such degradation, while others do
>  (Table 1, bottom block, FC100 + Mini-Imagenet; the one pointed out by the reviewer).
> This makes sense because in the domain-incremental setting where the output layer is shared between the two tasks, “similar” tasks are inherently more confusing (e.g., in certain sequences, FC100 label 1 may be similar to Mini-ImageNet label 3; while Omniglot examples are consistently very distinguishable from Mini-ImageNet examples).
>
> We also would like to emphasize that our main claim/focus is that our method combats “catastrophic” forgetting; we believe this is clearly demonstrated in our experiments compared to the baseline case without our ACL loss.
>
> > The results in Figure 2 are very confusing. Firstly, for Figure 2(a), given that meta-training is performed on the continual learning sequence consisting of two tasks, why is it also said that two tasks are learned simultaneously?
>
> > (**Requested changes**) Could you clarify the experimental setup and results analysis of Figure 2?
>
> Thank you very much for pointing this out; we should have written “meta-learned” instead of “learned” in the second paragraph of Sec. 4.2 and in the caption of Figure 2. We will correct this in the final version.
>
> With this correction, we hope all the concerns are clarified: the model has to “metalearn” to learn two tasks in a sequence. This requires acquiring 4 “sub-abilities”: being able to meta-learn a task presented at position 1 or 2, where the task is constructed from dataset A or B.
> In principle, there is no guarantee that being able to meta-learn task A at position 1 implies its ability to meta-learn task A at position 2, OR being able to meta-learn a task A at position 1 implies its ability to meta-learn task B at position 1, etc.
> Some of these **metalearning abilities** can emerge **simultaneously** during meta-training, while the actual “learning” of the two tasks is always **sequential**.
>
> To conclude, we thank the reviewer once again for their valuable feedback; your suggestions are very helpful for us to further improve our manuscript.

---

### Author Response · Authors · 2024-12-24
**End of discussion period is imminent & Revision uploaded**

Since we did not receive any response from the reviewers on our rebuttal, we currently assume there are no strong objections to our proposal regarding the resolution of the concerns raised (please let us know if this is not the case). We have updated our manuscript accordingly and uploaded the revised PDF. The main changes are highlighted in violet, and the list of changes can be found above.

We sincerely thank the reviewers once again for reviewing our work, and we wish you a wonderful holiday season.

---

### Decision · Action_Editor_GUkr · 2025-01-28

**Recommendation:** Accept as is

**Comment:**

The paper operates in a continual learning setting and proposes Automated Continual Learning (ACL), which meta-learns a self-referential neural network to mitigate a phenomenon they call 'in-context' catastrophic forgetting. All reviewers and myself agree that the paper meets standard for acceptance. Initial reservations focused on clarity regarding the meta-learning setup, the in-context catastrophic forgetting phenomenon, and the novelty of the approach. The authors’ revisions and responses sufficiently resolved these issues.

**Audience:**

Audience interested in continual learning, meta-learning / fast adaptation methods and self-referential NNs.

**Claims And Evidence:**

Yes, the authors have sufficiently addressed all reviewer concerns and provided clarifications on experimental details and setup.

---

> ### Author Response · Authors · 2025-02-17
> **Thank you and camera-ready uploaded**
>
> Dear Action Editor and Reviewers,
>
> Thank you for your favorable final decision. We have uploaded the camera-ready version.
>
> We sincerely appreciate the time and effort you invested in handling and reviewing our paper. Thank you very much once again.
>
> Best regards,
>
> Authors